# 3D GIS Platform for Flood Wargame: A Case Study of New Taipei City, Taiwan

Wen-Ray Su [1], Yong-Jun Lin [2,*] , Chun-Hung Huang [1], Chun-Hung Yang [1] and Yuan-Fan Tsai [3]

1   National Science and Technology Center for Disaster Reduction, 9F., No. 200, Section 3, Beisin Road, Xindian District, New Taipei City 23143, Taiwan; wrsu@ncdr.nat.gov.tw (W.-R.S.); chh@ncdr.nat.gov.tw (C.-H.H.); chyang@ncdr.nat.gov.tw (C.-H.Y.)

2   Center for Weather Climate and Disaster Research, National Taiwan University, No. 1, Section 4, Roosevelt Road, Daan District, Taipei 10617, Taiwan

3   Department of Social and Regional Development, National Taipei University of Education, No. 134, Section 2, Heping E. Road, Daan District, Taipei 10671, Taiwan; tyf@tea.ntue.edu.tw

*   Correspondence: vovman@gmail.com; Tel.: +886-3366-2614

**Abstract:** Wargames have been promoted by local governments in Taiwan since 2009, as they require far fewer resources than full-scale exercises. Previously, wargame scenarios were divulged before their launch, enabling participants to formulate response plans in advance, which made them ineffective. Currently, wargames in which scenarios are not shared in advance require a common platform for debriefing players regarding their planned actions. Owing to its geographical location, Taiwan is prone to significant flooding disasters. To assist in making countermeasures, we created a 3D GIS-based Flood Wargame Assistance Platform (FWGAP) for conducting rapid spatial analyses. Flooded areas are estimated in the FWGAP in three ways: (1) using a digital terrain model (DTM) with designated flood center and depth; (2) applying historical flooding spots; and (3) potential flooding maps. A FWGAP can estimate affected and vulnerable populations and has functions for locating resources such as shelters and hospitals near the flooded areas. Its integrated use of closed-circuit televisions, Google Street View maps, and 3D buildings to display flooded areas realistically ensures greater fidelity. This study reports on the city- and the district-level applications of the FWGAP. The results of the survey undertaken indicates that 56% of the participants agreed that the FWGAP enables disaster relief resources to be located on a GIS map. About half of the participants believed that a no-script flooding wargame using a FWGAP could help to identify problems in standard operation procedures and promote greater horizontal coordination among departments.

**Keywords:** disaster prevention response; decision support; geological information system (GIS) flood; wargame; platform; 3D



## 1. Introduction

Taiwan's geographical location makes it prone to frequent typhoons and heavy rains. Consequently, the government conducts annual disaster prevention exercises before May, which marks the start of the flood season. A wargame on disaster prevention is one of the exercises that local governments can deploy, with its primary objectives being to strengthen the knowledge and skills of disaster prevention personnel in the following areas: disaster-prevention standard operating procedures (SOPs), creating inventories of resources required to deal with disasters, and identifying potential problems that could be encountered during disasters.

The wargame (the German- "Kriegsspiel") was invented in 1812 by von Reisswitz and is used to simulate military operations [1].

The US Federal Emergency Management Agency (FEMA), Department of Homeland Security (DHS), defines seven types of emergency exercises: full-scale exercises, functional exercises, drills, games, tabletop exercises, workshops, and seminars [2]. The first three

exercises are operations-based, and the others are discussion-based. Disaster wargames belong to the games and tabletop exercises. In 2004, FEMA conducted the "Hurricane Pam" tabletop exercise with scenarios of heavy rain in southeast Louisiana and a storm surge topping levees in the New Orleans [3]. It launched before Hurricane Katrina in 2005 and identified the need for approximately 1000 shelters [4].

The UK government has delineated four categories of emergency exercises: discussion-based, tabletop, live, and combinations of the above [5].

In Japan, the Disaster Imagination Game (DIG), which is a tabletop exercise, was created in 1977 to facilitate earthquake preparation. Since then, the game has been extended to cover different types of hazards, such as flooding and debris flows. Participants in the game can draw hazard hotspots and resources on maps and discuss possible strategies to be deployed [6]. Recently, the DIG has been extended to include the safety of households, enabling family members to discuss safety issues in home-related disaster prevention plans, such as unmounted furniture that may move around, thus injuring family members during an earthquake [7].

In recent years, Taiwan has adopted several disaster wargames. The NFA (2012) [8] has identified the following three main categories of disaster wargames: single emergency operation centers (EOCs), multiple EOCs in different locations, and multiple EOCs in the same location. The Homeland Security Office (HSO) [9] has placed emphasis on the importance of the wargame platform for assisting in the exercise. However, the HSO refers to those boards with maps on them, which are adopted in military services or hard-copy, printed maps.

Many games on disaster prevention have been developed, some of which are computer-based while others are card or activity games. Some adopt imaginary flooding scenarios, and some adopt the results of hydraulic modeling. The former focuses more on the educational purposes, which is more suitable for students. The concept of flood protection is more critical. The latter focuses more on the training of the disaster personnel and so the fidelity of the scenario is essential.

## 1.1. Educational Flooding Wargame

Some educational flood games are described as the following. Tsai et al. [10,11] developed a flood protection game with a 2D interface, and they demonstrated its effectiveness in increasing student motivation levels to participate in a flood protection campaign. Tsai et al. [12] also developed the "Battle of Flooding Protection" game to promote student education on disaster prevention. This game, which has a 3D interface, enhances student skills, motivation, self-awareness, and civic responsibility for flood protection. However, the scenarios in these educational games are imaginary.

FloodSim is an online policy simulation game that focuses on adopting structural and non-structural measures to minimize the risk of flooding when there are financial constraints [13]. It is very similar to Tasi's forementioned games. The game features three-dimensional buildings and the accompanying flooded area.

UNDRR launched a web-based disaster education game called Stop Disasters (https://www.stopdisastersgame.org/ accessed date: 1 August 2021). In this game, two-dimensional tiles represent land areas, and measures can be taken to accommodate users or protect them from the disaster's effects within a specific budget. The hazards encountered within the game include tsunamis, hurricanes, wildfires, earthquakes and floods. Once the users have completed their individual constructions, they can press the "start disaster" button to initiate a disaster and obtain a detailed report about damaged houses and the number of dead and injured persons [14].

FloodSim and Stop Disasters are computer-aided games, but the maps used in these games are conceptual. The flooding images in FloodSim are remarkably realistic when the three-dimensional buildings are overlaid with, for example, flooding of the nearby River Thames in London. However, none of the games are GIS-based, but such maps would

provide participants with a more immersive experience. Those games aim at educating the students and the public.

### 1.2. Flooding Wargame System Using Modeling Scenarios

As for disaster prevention wargame systems using 2D hydraulic modeling results, Lee et al. [15] developed a two-dimensional hydraulic model to simulate historical and future flooding events in Xizhi, New Taipei City, Taiwan. This web-based system provides participants with the required maps and interactive scenarios. The flooding scenarios that are generated are based on the quantity of rainfall, and although the model depicts the possible flooded area, it is not GIS-based. Chen et al. [16] evaluated the effectiveness of a wargame platform regarding the emergency evacuation of residents during a flood. This platform, which integrates qualitative and quantitative methods, aims to show how the responsible authorities take measures to deal with a flooded area based on a historic flood event. The authors concluded that the system could improve the flexibility and proficiency of those in charge and aid in the training of the disaster prevention personnel.

### 1.3. Other Flood Game Tools

To increase the fidelity of the flooding wargame, different techniques were used. Tsai [17] introduced an interactive electronic whiteboard as a wargame platform to evaluate the effectiveness of disaster prevention in Minxiong Township, Chiayi, Taiwan. Hsieh [18] proposed a reality-based interactive war simulation platform using a Microsoft Kinect 3D camera and projector. Researchers at Coventry University developed two strategy-level exercises using a Second Life VR platform, one of which was for humanitarian aid, and the other for a hydroelectric dam exercise [19]. In addition, they developed an effective exercise for flood emergency response planning using the OLIVE VR platform [20].

Tsai et al. [21] developed a drill script generator for compound natural disasters, including typhoons, earthquakes, and debris flows based on historical events. The location of the generated script is shown as a dot on a hyperlinked map.

### 1.4. Comparison of Flooding Games

Table 1 shows the functions of the different computer-assisted, flood-related disaster games. The GIS-based Flood Wargame Assistance Platform (FWGAP) developed for this study is listed for comparison. As shown in Table 1, some games use VR platforms, such as the two games created at Coventry University [19,20] and the one created by Hsieh [18]. The former, constructs a virtual world for the participants using avatars but does not require headsets. The OLIVE platform provides three-dimensional Google Earth maps to aid participants, whereas the format of the Second Life platform resembles that used in educational courses in which students discuss the best strategy for building a dam. Heish's game entails the use of a Microsoft Kinect camera for interactive actions.

The games developed by Tasi et al. [10–12], FloodSim, and Stop Disasters, are educational, and their objective is to minimize the risk of flooding due to financial constraints. Like chess, these games aim to win wars, but their ultimate goals are to reduce the effects of flooding. Tasi et al. [18] has the only script-generating function among the others in Table 1.

Of the 11 flood wargame-related literature in Table 1, three use a GIS-based platform. Fang [22] noted that the GIS-based wargame system, involving integrated overlaid theme maps, represents the future trend. Four platforms (FWGAP, FloodSim, and those developed by Lee et al. [15] and Chen et al. [16]) use two-dimensional overland flow simulation maps, and three platforms (FWGAP, FloodSim, and the one developed by Tasi et al. [11]) use three-dimensional buildings, which make the flooding appear more realistic. In addition, the FWGAP is the only game that includes a spatial analysis function, enabling the participant to look up data and display it quickly on a GIS platform.

*1.5. Flooded Area Assessment Using Elevation Data*

A volume of techniques was proposed to assess flooded areas using digital elevation data (DEM). Manfreda et al. [23] adopted a modified topographic index using DEM for detecting flood-prone areas. This method takes into account the drained area and local gradient. Samela et al. [24] adopted hierarchical filling-and-spilling based on DEM for pluvial flood hazard assessment. The proposed method can handle the flow connectivity among low-lying areas such as pits. In other words, when the water surface level is high enough and then overflows to another pit. In addition, the rainfall fills every pit at the same time, and is termed hierarchical. Those methods can be regarded as simplified, distributed hydrological modeling. They take some time to compute, although they are less time-consuming than the typical 2D hydraulic modeling.

The high resolution, sub-grid model [25] is widely adopted in the new-generation 2D flow model, such as the USACE HEC-RAS 2D model [26]. For example, a model can be built from a detailed terrain model (5 m × 5 m) grid, with a computation cell size of 20 m × 20 m. The water tends to flow to low-lying areas due to gravity. The cell can be wet with the correct water volume in the sub-grid model with a given water surface level. The model established the elevation-volume relationship. We adopt this simple technique by designating the center of the flood as well as the flood depth. The water surface is then horizontally extended to nearby digital terrain grids. We achieved an acceptable result when comparing to 2D hydraulic modeling. The time for this method takes less than one minute.

Green [27], suggested that wargame exercises could be used to train participants and enhance their understanding of a disaster preparedness plan, the allocation of resources, and the assignment of tasks. Full-scale and functional exercises are much costlier than games or tabletop exercises, and they are conducted less frequently [28]. In short, a wargame can be launched as needed and consumes fewer resources when compared with other exercises.

Starting from 2009, Taiwan has promoted the use of disaster wargames by local governments with the aim of identifying potential problems. Initially, the scenarios were accessible to the participants, enabling them to develop response plans in advance. To avoid possible errors in the wargame that could be discovered by supervisors, the participants drafted scripts for the scenarios before commencing the game. During this period, the participants followed and vocalized the script and did not interact with each other.

Lovelock [29] proposed that a simulation should entail eight elements, one of which is, that "games should contain an element of surprise." However, in the case above, because the participants acquired the scenarios before participating in the wargame, there was no element of surprise. The director of the corresponding team in the wargame served merely as an observer and did not attempt to develop coordination among the team in responding to the scenarios. This kind of wargame was evidently time consuming and lacked practical value.

Currently, to improve the effectiveness of the wargame, its scenarios are not provided in advance. No scripts are prepared beforehand, and the team has to respond to situations during the exercise. Thus, although the problem of drafting the script in advance has been solved, the observers cannot visualize what the participants had planned in the absence of spatial information. For this reason, the participants need a common platform to perform a debriefing of their planned actions.

**Table 1.** The functions of different computer-aided, flood-related disaster games.

| No. | Name | Flood Area Estimation | 2D Overland Flow Simulation Maps | Historical Flooding Area | Affected Population Estimation | Nearby Evacuation Shelter | Nearby Fire Brigade | Nearby Elderly Care Center | Nearby Hospitals | 3D Google Earth Maps | Google Street View | CCTV | GIS | 3D Building | VR | Script Generator | Educational |
|---|---|---|---|---|---|---|---|---|---|---|---|---|---|---|---|---|---|
| 1 | FWGAP (this study) | v | v | v | v | v | v | v | v | | v | v | v | v | v | | |
| 2 | Lee et al. (2003) | | v | | | | | | | | | | v | | | | |
| 3 | Large flood emergency response planning exercise Chen et al. (2013) | | v | | | | | | | v | | | | | v | | |
| 4 | Hsieh (2013) | | | | | | | | | | | | | | v | | |
| 5 | Hydroelectric dam exercise Chen et al. (2014) | | | | | | | | | | | | | | v | | v |
| 6 | Tasi et al. (2014) | | | | | | | | | | | | v | | | v | |
| 7 | Tasi et al. (2015a; 2015b) | | | | | | | | | | | | | | | | v |
| 8 | Chen et al. (2016) | | | v | | | | | | | | | | | | | |
| 9 | FloodSim (Rebolledo-Mendez et al., 2008) | | | | | | | | | | | | | v | | | v |
| 10 | Stop Disasters (UNDRR, 2019) | v | | v | | | | | | | | | | | | | v |
| 11 | Tasi et al. (2020) | | | | | | | | | | | | | v | | | v |

To address this need, we propose the use of the FWGAP for flooding-related wargames. Spatial analysis can be rapidly performed in the FWGAP for use by the response team, and the flooded area can be estimated, thereby facilitating the exercise.

Participants using the FWGAP can access the impacts of the flooding, such as the area flooded, the affected population, and the presence of any centers in the flooded area that care for the elderly.

Three methods within this platform are used to estimate the extent of the flooded area: assigning the flooding center and the depth, determining historical flooding spots, and using a flooding depth map based on rainfall scenarios. In addition, the FWGAP is able to perform spatial analysis to determine the affected population and nearby centers caring for the elderly. The disaster reduction taskforces or resources needed for disaster mitigation, such as a fire department, police, hospitals, and evacuation shelters, can also be displayed on the platform (Table 1). The participants can then use that information to implement feasible measures, such as sending the evacuees to unaffected (non-flooded) shelters according to their capacities and locations. In the past, the participants would bring tables containing a list of the resources and look them up. The FWGAP speeds up the process of acquiring all of the relevant data, which are stored in the database. Being a web-based GIS platform, new functions can be easily added to FWGAP.

Immersion techniques or games enable participants in emergency management training programs to acquire the knowledge and skills needed for tackling the different phases of the emergency [30]. Computer-aided exercises can increase the degree of the immersion of the participants in the scenario. To achieve the goal of providing participants with such an immersive experience and displaying the flooded areas realistically, closed-circuit televisions (CCTVs), Google Street View maps, and three-dimensional buildings are integrated into the FWGAP. Thus, participants do not need to go to the field; they can use these tools and see real-time CCTV images.

We administered a questionnaire for participants to evaluate the FWGAP's performance and the platform's fidelity, functionality, and ease of use. This study illustrates applications of this system in New Taipei City, Taiwan, and its districts, while also reporting on the feedback elicited from users. Some functions recommended by the users have been incorporated into an upgraded version of the FWGAP.

## 2. Methods

The National Science and Technology Center for Disaster Reduction (NCDR) in Taiwan developed the Disaster Response Decision Support System (DRDSS) in 2010, which has since been applied in many natural disasters. DRDSS users have tested the system, which enables the overlaying of maps and contains spatial analysis tools that facilitate the decision maker's grasp of the trajectory of events. Moreover, the DRDSS provided accurate information that could be used by the director of the Central Emergency Operation Center (CEOC) to select appropriate countermeasures.

In 2016, the FWGAP was launched within the DRDSS to respond to the needs of personnel dealing with disasters. The FWGAP builds on the DRDSS and provides the responsible personnel with an opportunity to familiarize themselves with potential flooding spots, the number of affected persons, and available resources. The methods adopted in the FWGAP are described below.

*The Flood Wargame Assistance Platform (FWGAP)*

When decision makers are under time constraints, they tend to reduce losses through risk avoidance in cases where negative benefits are anticipated [31,32]. Disaster response decisions concerning the security of lives and property means decision-makers are often under pressure to make the right decisions [33]. In other words, if people make wrong decisions, they generally blame the decision makers and sometimes demand compensation for their losses.

The conceptualization of disaster prevention has shifted from simply implementing countermeasures during disasters, to pre-disaster planning and disaster management. Therefore, disaster prevention personnel require more predictive information on disasters to provide them with more time to take appropriate measures, such as evacuation. The presence of an early warning system combined with disaster potential analysis facilitates the evacuation of people in disaster-prone areas prior to the occurrence of a disaster, thereby reducing losses in lives and property [34]. In short, the concept of "disaster prevention" has changed to "disaster reduction."

Making the right decisions is critical in disaster management, and accurate and timely information is essential for making such decisions. The DRDSS provides integrated tools that enable access to precise and timely information gathered from different agencies and platforms. For example, weather information is sourced from the CWB, and information on debris flow is obtained from the Soil and Water Conservation Bureau (SWCB). Familiarization with the exercises enables the trainees to deal effectively with similar cases. In other words, habituation, especially under pressure, leads to better performance [35]. This platform mainly provides assessments of disaster effects and resources in flood-prone and low-lying areas, to enable disaster prevention personnel to familiarize themselves with the decision-making operations related to emergency events.

1. Analysis of user needs.

The following suggestions were provided by surveyed respondents:

(1) Assessments of flooded areas:

Flooded areas can be delineated based on the locations of historical disasters, spatial analysis, or simulations using flooding models.

(2) Assessments of affected populations.

When a disaster occurs, in addition to determining the affected area, the disaster relief personnel must also determine the affected populations to organize evacuations.

(3) Estimations of vulnerable residents.

Residents of high-rise buildings do not need to be evacuated. However, disabled or frail individuals on the first floors of such buildings located in low-lying areas should leave first, as they belong to the category of vulnerable people. This category includes both disabled individuals and older adults above 65 years of age.

(4) Selecting evacuation shelters.

When floods occur, people living in low-lying or flood-prone areas may need to be evacuated to shelters. Therefore, the issue of the manner in which evacuation shelters, which should be safe from flooding, are selected is important. Their selection can be facilitated using the platform's spatial analysis tools. Appropriate evacuation shelters can be referenced in the area extending from the flooding impact radius to the center of the circle.

(5) Assessments of disaster relief resources.

When individuals in the affected areas need to be evacuated and the appropriate evacuation shelter has been selected, the resources required to carry out the evacuation can also be displayed on the platform. Disaster relief resources in neighboring areas, such as fire brigades, police stations, and hospitals, must be listed for reference purposes.

We designed the FWGAP to include these tools.

2. Function Development

Simplicity is emphasized in the functional design of the FWGAP. After the user has entered the settings for the flooding location and depth, the flooded area, affected population, and number of vulnerable people are automatically calculated. By calculating the number of evacuation shelters near the flooded area and the number of people evacuated

from the affected area to the designated evacuation shelter, the FWGAP facilitates decision making in respect of the flow of evacuees to the shelters.

The FWGAP's map display interface has been developed using the ESRI ArcGIS API suite of products. The Digital Terrain Model (DTM) grid data contains more than 45 million grid data points. If it were to be carried out using an ordinary database, the computation would take too long, which would affect the display's performance. In view of the vast amount of data, we used the ESRI and CARTO spatial processing service to accelerate the spatial analysis and display efficiency.

The FWGAP has a web-based interface (Figure 1), with tools for estimating the flooded area, affected population, vulnerable people, and for locating nearby shelters. Users can switch functions using the tabs on the tab bar and can use the tabs to access the theme map quickly. Figure 2 depicts a flow chart of the tab functions and methods applied in the FWGAP, as described below.

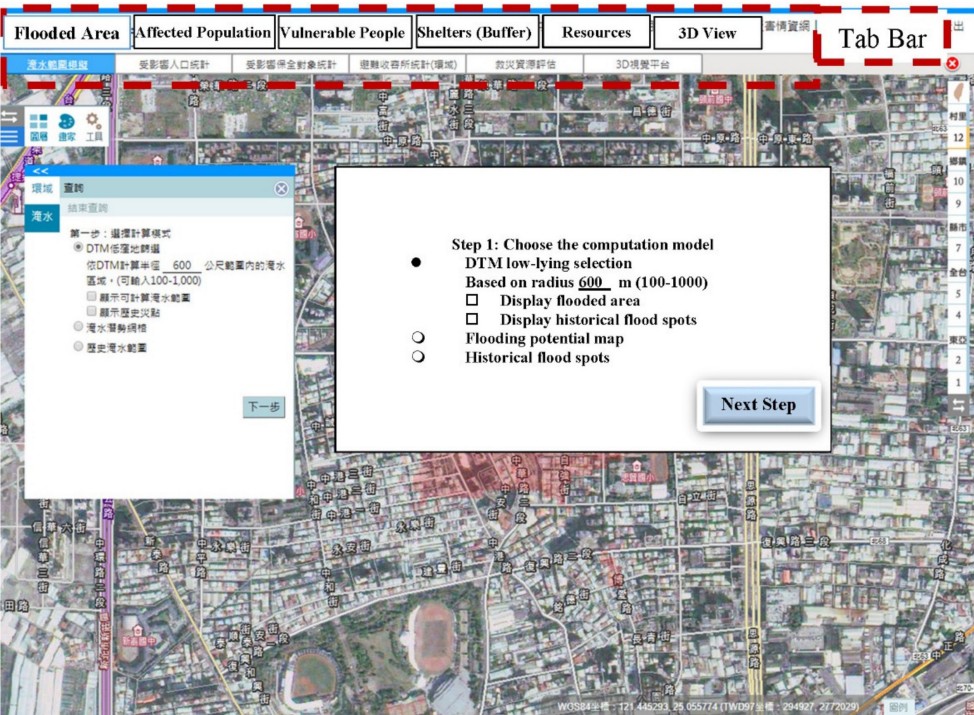

**Figure 1.** Function tabs in the FWGAP.

(1)    Tab 1: Calculation of the flooded area.

The following three methods can be used to estimate the flooded grids.
**Method 1**: calculating the center point and flood depth.

The center of the flooded area is set first, followed by the flooding depth (e.g., 0.5 m or 1 m). Next, geoprocessing is performed within the FWGAP using CARTO. Grids that meet the established criteria within the radius of the flooded area are exemplified (Figure 3). The elevation of the center point adds the flooding depth equal to the water surface level (WSL). The WSL is horizontal and extends to the nearby grids shown in the DTM. There are three ways of specifying the center point: clicking on the map, entering the building's address, or determining the coordinates. The calculation results are depicted in Figure 4 (the red areas) and serve as the basis for other functions.

The hypothesis of this method assumed that floodwater flows into the low-lying area due to gravity. Therefore, the water surface level would be horizontal. When encountering large floods in bowl-like, low-lying areas (such as New Orleans in Hurricane Katrina in 2005), this is usually true. However, in this case, the drainage was full, and the excess water may be deposited in the low-lying area. Therefore, we adopted the concept of a sub-grid [25,26] to assess flooded areas quickly.

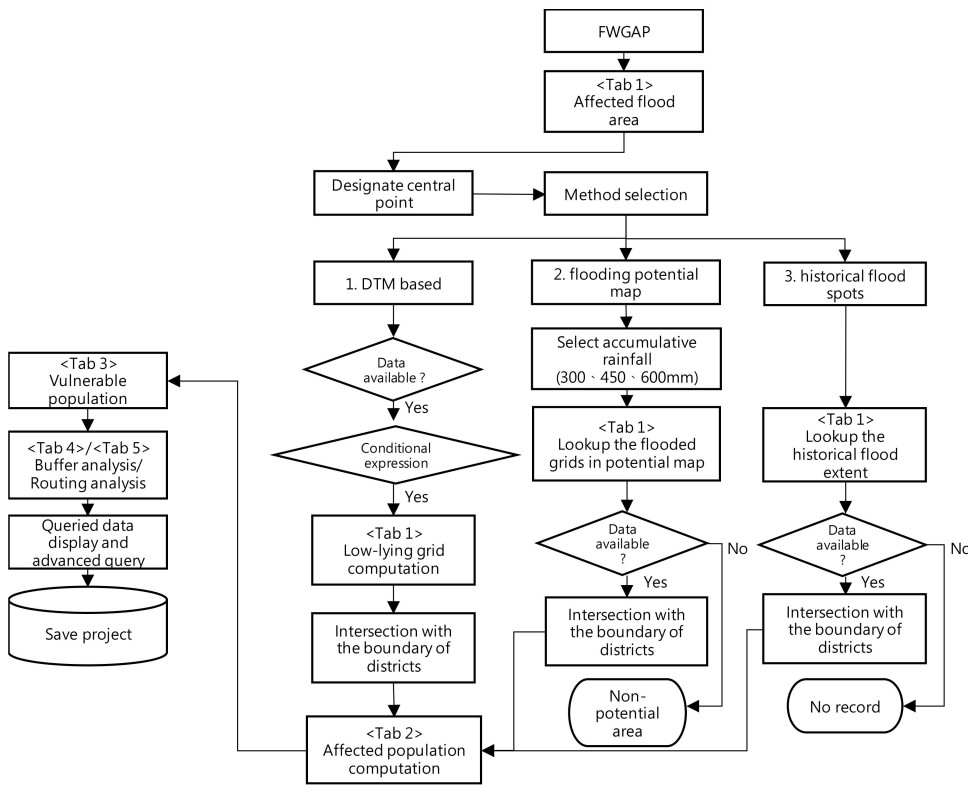

**Figure 2.** Flow chart of steps in the FWGAP.

For comparison, a historical flood event from 2018 using a 2D hydraulic model is shown in Figure 5b, and the results of the FWGAP are shown in the Figure 5a. The maximum flood depth of the historical event was between 1.0–2.0 (m) with total rainfall of 572 (mm) and peak hourly rainfall of 108 (mm/h), which exceeded the drainage capacity of 60–70 (mm/h). The flooding extent of the two was similar. It can be said that the flooding extent of the FWGAP was larger than that of the 2D hydraulic model. The FWGAP set a flood depth of 1.5 (m), with the center shown in a figure in Figure 5a. Some of the flooding extents of FWGAP were located in the floodplain. The deepest flooded areas were similar, but the 2D hydraulic model was more realistic in the lower left part of the simulation domain. The overlaid flooded grids of the two divided by the total flooded grid of FWGAP is 74%. Among other cases, the ratio mentioned above is about 62~74%.

In reality, the floodwater may run very fast and has a gradient. When water is not horizontal, it flows due to gravity until the water surface level is horizontal. Moreover, FWGAP cannot simulate the flood induced by levee breaching, overtopping the levee, etc.

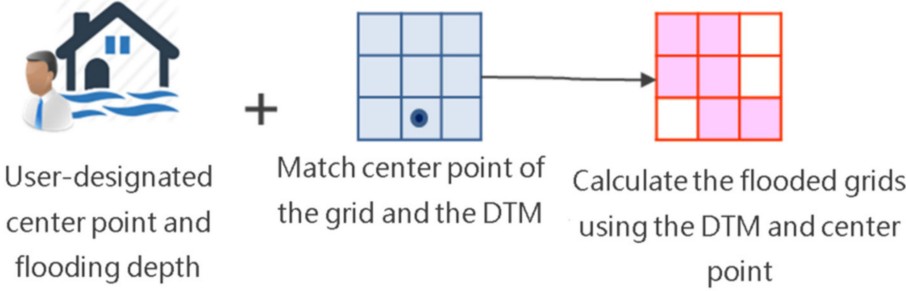

**Figure 3.** A depiction of the flooded grids concept.

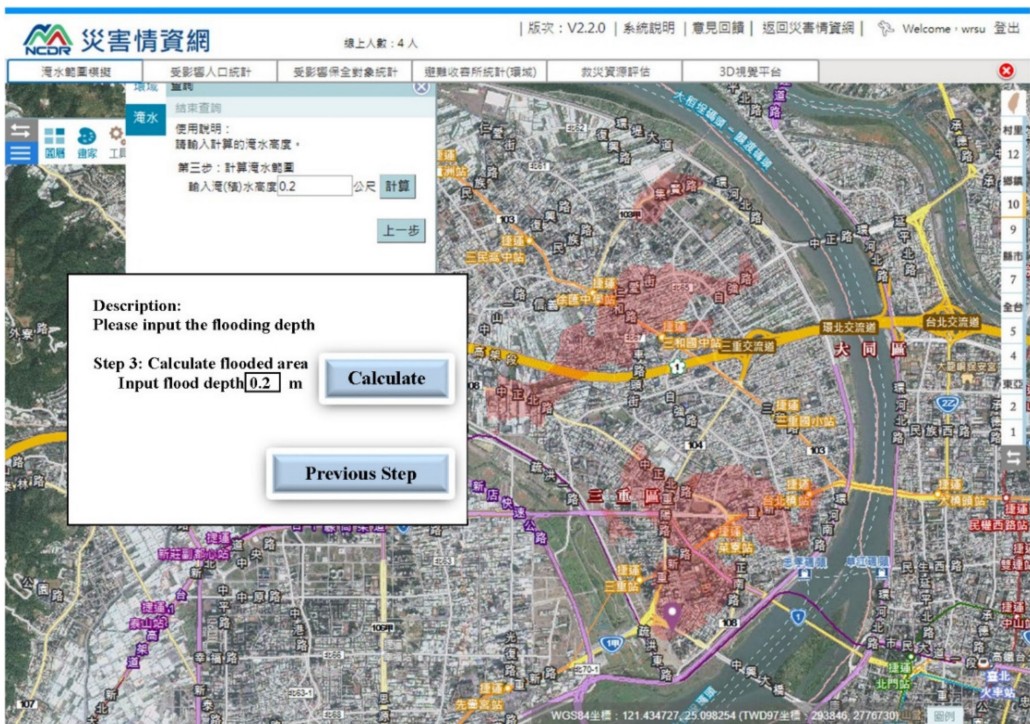

**Figure 4.** Method 1: Using the center point and flooding depth to calculate flooded areas.

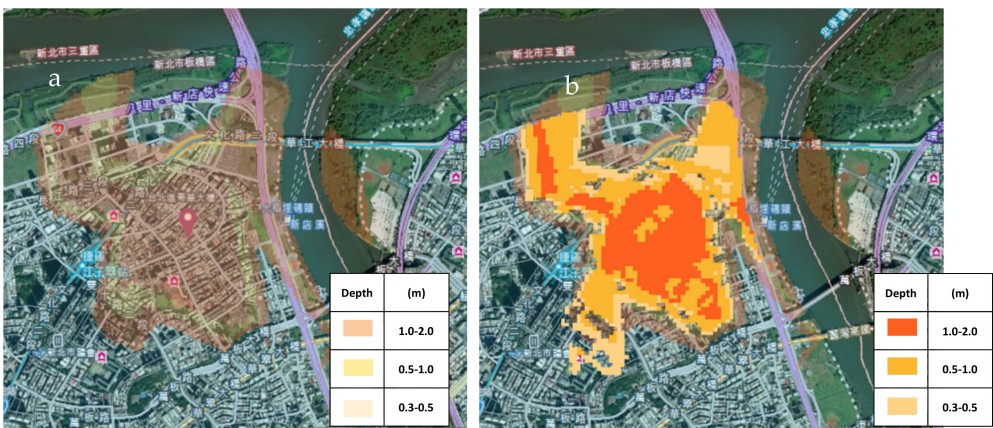

**Figure 5.** Comparison between (**a**) FGWAP and (**b**) 2D hydraulic modeling overlaid FWGAP.

**Method 2**: historical flooding spots

Natural or anthropogenic factors can induce flooding events. However, floods will not recur in the future if their causal factors no longer exist. For example, flooding will not occur if the drainage system has been improved. Therefore, to grasp the actual flooding situation fully, data on flood events that have occurred in New Taipei City over the past five years were compiled and a flooding polygon of the events was mapped using a GIS mapping service. Users of the FWGAP can click on the "historical flooding spots" tab in the interface to input these data into the analysis. This function can be used to screen out the extent of historical flood events, as shown in Figure 6 (the pink polygon around the purple landmark sign).

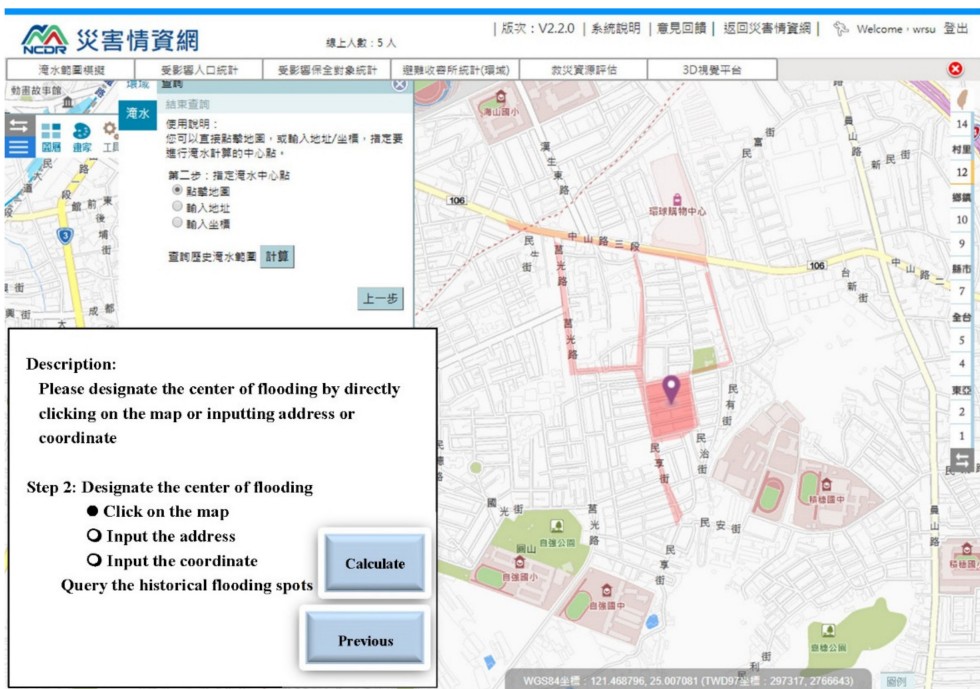

**Figure 6.** Method 2: Using historical flooding spots to calculate flooded areas (depicted as a pink polygon surrounding a purple landmark sign).

**Method 3**: constructing a map of the potential flooded area.

First, the center point was specified, followed by the selection of the potential flooding situation, such as the flooding potential of 600 mm of daily rainfall (Figure 7). The flooding potential map uses the maximum flooding depth produced with the SOBEK two-dimensional flooding model [36]. The SOBEK model uses the following two-dimensional overland flow mass and momentum equations:

Mass conservation:

$$\frac{\partial h}{\partial t} + \frac{\partial (ud)}{\partial x} + \frac{\partial (vd)}{\partial y} = q_{lat} \tag{1}$$

Momentum conservation:

$$\frac{\partial u}{\partial t} + u\frac{\partial u}{\partial x} + v\frac{\partial v}{\partial y} + g\frac{\partial h}{\partial x} + g\frac{u|V|}{C^2 d} + au|u| = 0 \tag{2}$$

$$\frac{\partial v}{\partial t} + u\frac{\partial v}{\partial x} + v\frac{\partial v}{\partial y} + g\frac{\partial h}{\partial x} + g\frac{v|V|}{C^2 d} + av|v| = 0 \tag{3}$$

where $x$ and $y$ are horizontal coordinates in the study area; $u$ and $v$ are velocities in $x$ and $y$, respectively; $t$ is time; $d$ is the water depth; a is the wall friction roughness; g is the gravitational acceleration; $C$ is the Chezy roughness coefficient; $q_{lat}$ is the lateral flow; and $V = \sqrt{u^2 + v^2}$.

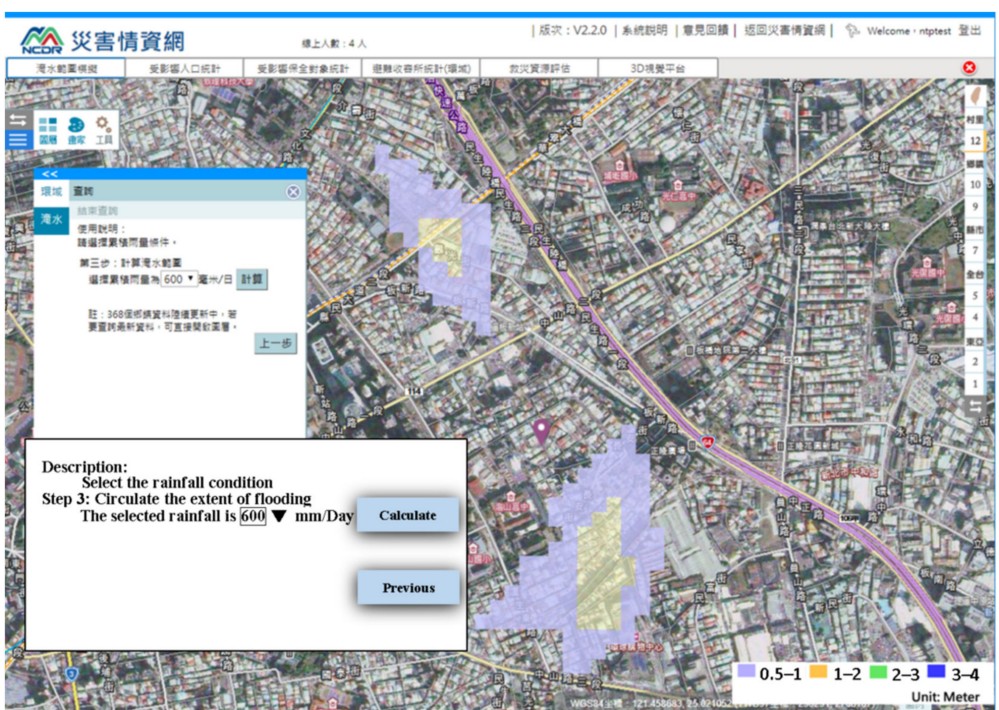

**Figure 7.** Method 3: Using a flooding potential map to calculate flooded areas.

(2)    Tab 2: Computation of the affected population.
        Following the spatial calculation of the flooded grid using Tab 1, the affected popula-
        tion data are filtered using the GP service, resulting in the number of affected people
        displayed on the map screen. The filtering criteria are based on population statistic
        units that are colored light green in Figure 8. In this example, the affected population
        comprised 7216 persons.

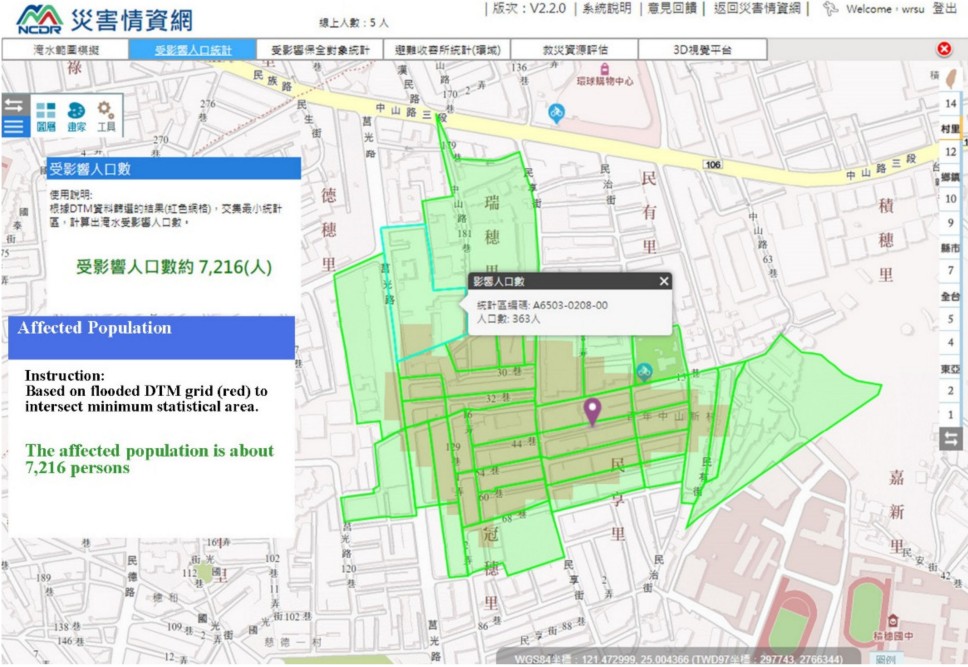

**Figure 8.** Tab 2: Estimating the flood-affected population.

(3) Tab 3: Estimating the vulnerable population.

The vulnerable population can be estimated using spatial analysis according to the flooded grids. Figure 9 shows the result of an illustrative scenario with 346 vulnerable persons, including elderly or disabled individuals.

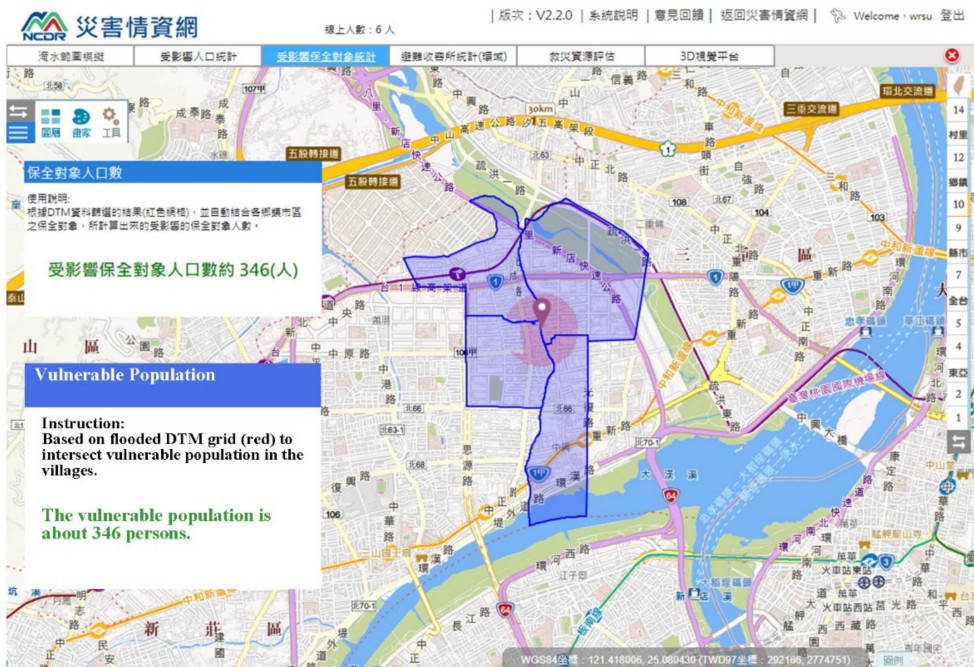

**Figure 9.** Tab 3: Estimating the vulnerable population (346 persons in this case study).

(4) Tab 4 and Tab 5: Buffer and routing analysis.

Following the estimates of the affected and vulnerable populations, the next task was to decide how to evacuate these people to nearby shelters. This was determined through point domain analysis, which can be performed using Tab 4 of the FWGAP, as shown in Figure 10. The system automatically used the center point set using Tab 1 and selected the shelters located within the specified radius, showing their respective capacities. The decision makers could then authorize the distribution of evacuees to the nearest shelter.

Tab 5 was used for the routing analysis and also served as a tool for screening shelters or other resources. This tab uses the ESRI routing analysis module to calculate the resources available within a traveling distance of 2.5 km. This distance measurement, based on the routing analysis, is depicted as a light blue dashed line in Figure 11. The absence of shelters within the delineated area indicates that the shelter was more than 2.5 km away.

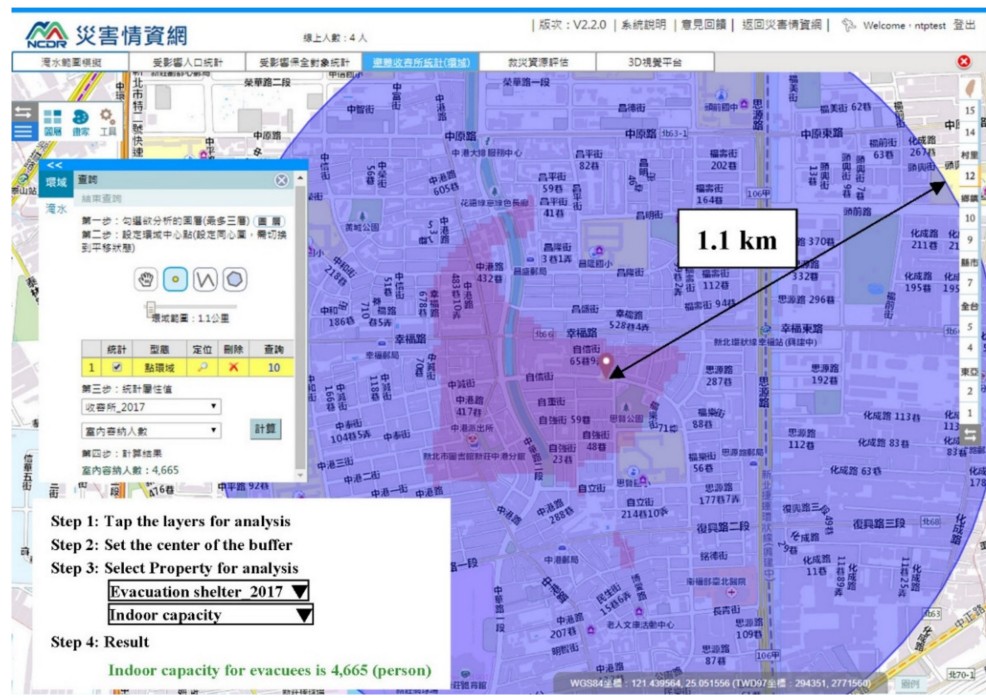

**Figure 10.** Tab 4: Buffer analysis to locate nearby resources and their capacities within a radius of 1.1 km (in this case study).

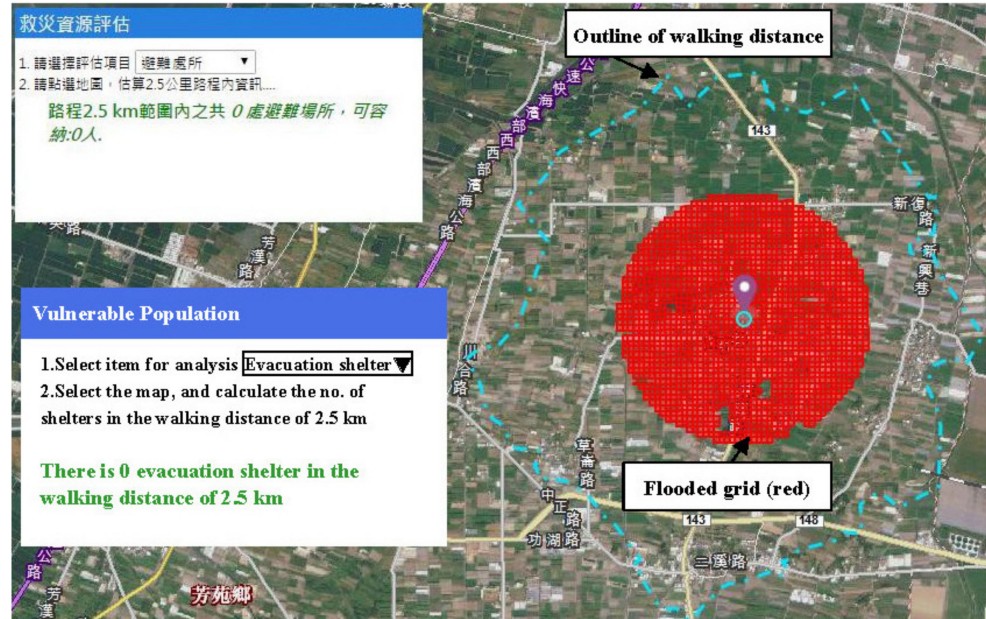

**Figure 11.** Tab 5: Routing analysis of nearby resources using an example of evacuation shelters (the walking distance is outlined using a light-blue dashed line).

### 3. Results-A Case Study of New Taipei City

The World Natural Disasters Hotspot Report released by the World Bank [37], identifies Taiwan as the place that is most vulnerable to natural disasters globally, irrespective of whether the assessment is based on the percentage area or population. With a population of about 4 million within 29 districts, New Taipei City is the most populated Taiwanese city. The FWGAP was applied at the city and district levels, as described below.

#### 3.1. City-Level Flood Wargame

A disaster wargame was launched on 31 May 2019 for the flood prevention authorities in New Taipei City. The scenario was based on a historical flooding event induced by frontal rainfall that occurred on 2 June 2017 and induced 540 flooding points in New Taipei City. The main flooded area was located near the northern coast and estuary of the Tamsui River. From 02:00 on 2 June 2017 to 07:00 on 3 June 2017, Sanzhi District had the most hourly rainfall (112 mm), followed by Tamsui (107.5 mm). The accumulated rainfall was greatest in Sanzhi District (707.5 mm), followed by Shimen District (677.5 mm).

In this city-level wargame, the FWGAP was applied in an assessment of the flood-affected area using the flooding depth setting. The affected population, nearby location of shelters, and social welfare institutions for the elderly were ascertained, to ensure appropriate measures were taken. In the past, a care center for the elderly was flooded during Typhoon Fanapi that hit southern Taiwan, resulting in the immersion of immobile elderly residents in floodwater [38]. The FWGAP displayed three-dimensional buildings, CCTV images, and Google Street View maps of the affected area. Below are some screenshots of the wargame from the FWGAP, along with brief descriptions.

1. Assessment of the flood impact in Sanzhi District

The flood depth was set at 0.5 m and the impact radius at 1 km in the FWGAP. Low-lying areas of Sanzhi District along Tamking Road were identified as the flood-affected area. The center of the affected area is marked in purple in the map (Figure 12) and the number of affected persons was 569 (Figure 13).

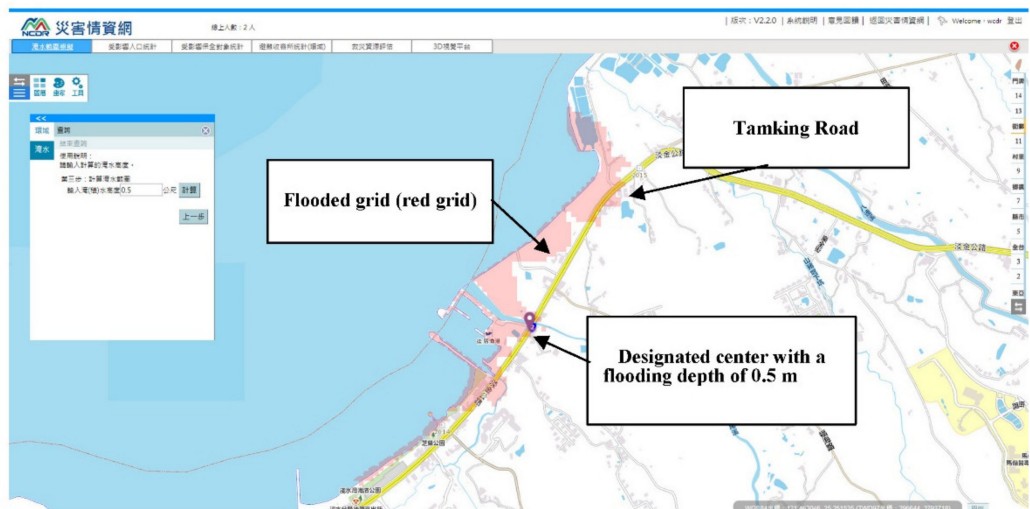

**Figure 12.** Flooded area of Sanzhi District.

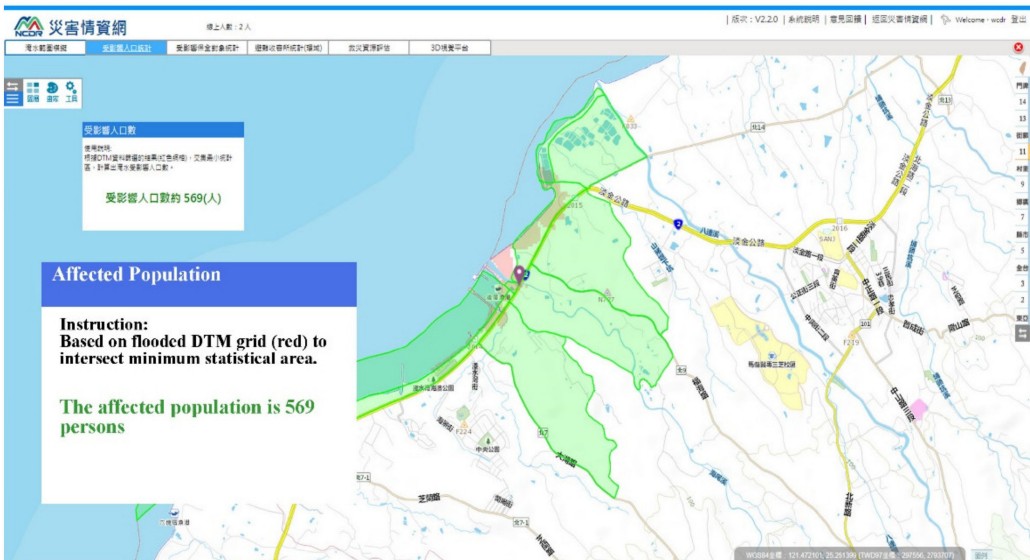

**Figure 13.** The flood-affected population of Sanzhi District.

2. Flood impact assessment of Sanchong District

The hourly rainfall in Sanchong District was 50 mm or more for three consecutive hours. The flooded area within the district, with an estimated radius of 100 m and a flooding depth of approximately 1.5 m, included Sanmin Street and Zhongzheng North Road (Figure 14). The affected population numbered 2927 persons. Figure 15 shows the estimated flooded area.

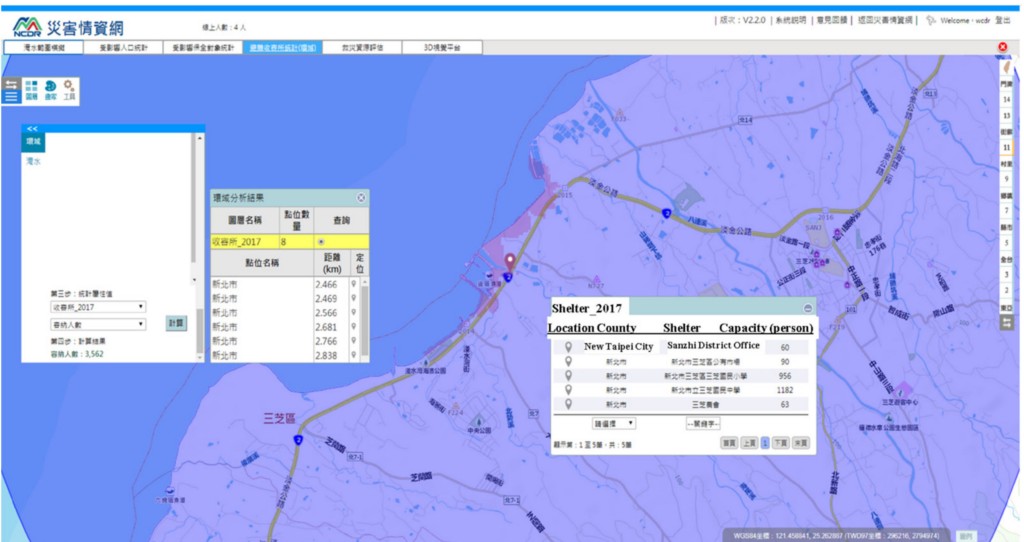

**Figure 14.** Buffer analysis of shelters in Sanzhi District within a 4-km buffer zone.

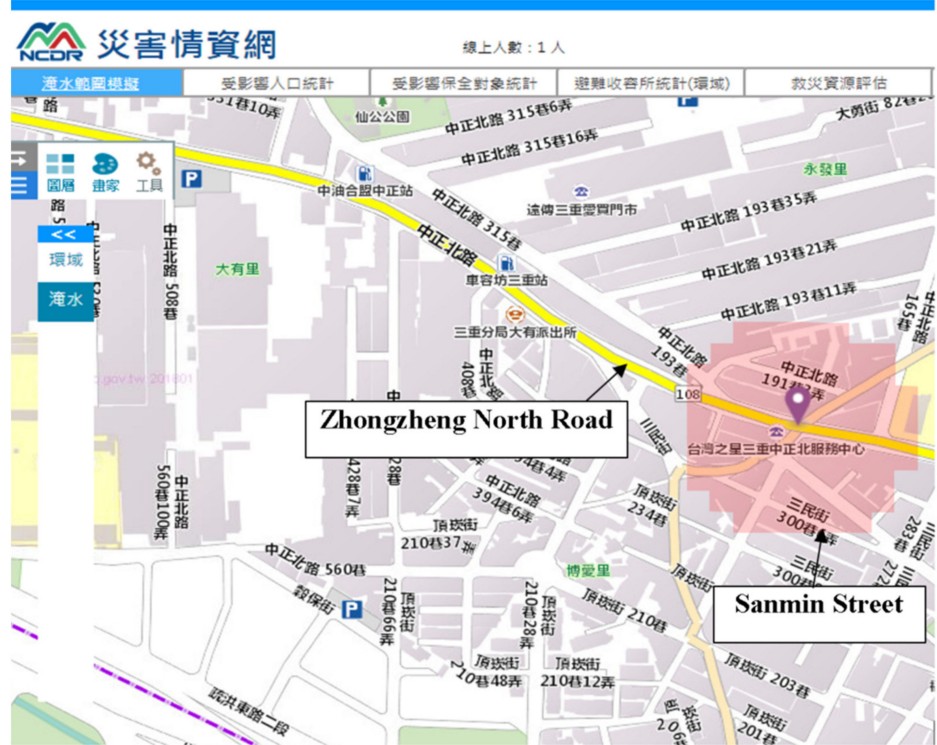

**Figure 15.** Flooded area in Sanchong District (depicted as a red grid).

3.     An assessment of the flood impacts in Wugu District

The hourly rainfall in Wugu District was 50 mm or more for three consecutive hours. The depiction of three-dimensional buildings and a Google Street View map facilitated the wargame participant's on-site observation of the flooded area. The yellow figure in the image to the left in Figure 16, indicates the center of the street view shown in the image to the right. A gas station can be seen in the image to the right in Figure 16, and the flooded area is shown to its left. There were 404 people in the affected area, and the nearby shelters were not affected by the flooding. Three police stations were located within 2.5 km of the flooded area, the nearest being Wugong Police Station. The relevant agencies in New Taipei City used the FWGAP to evaluate the area impacted by the flood, along with the affected population, disaster relief resources, and shelters in the vicinity (Figure 17).

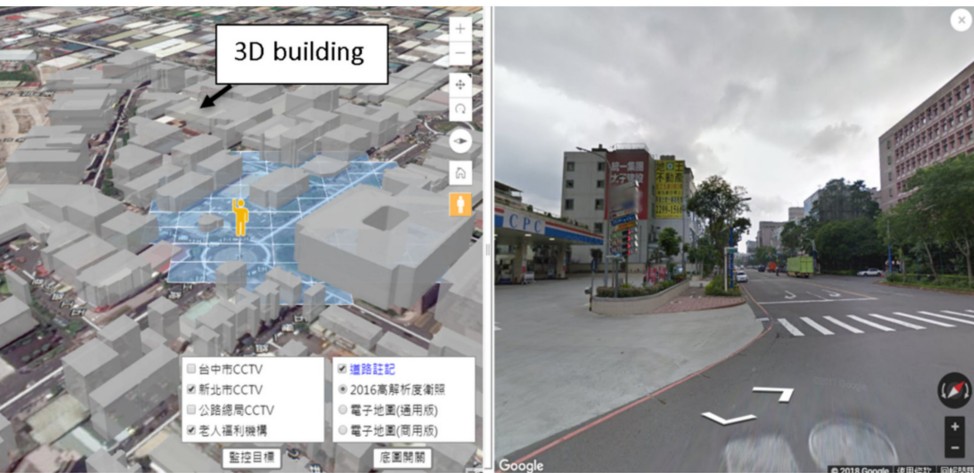

**Figure 16.** Three-dimensional buildings and a Google Street View map of Wugu District.

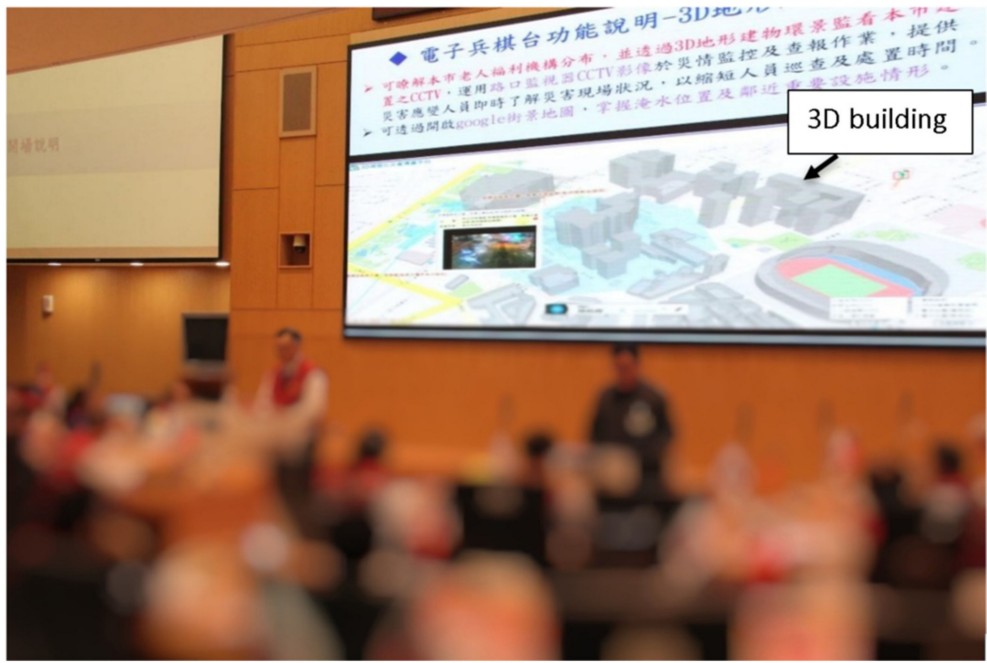

**Figure 17.** The responsible agencies in New Taipei City using the FWGAP—3D Building (on 31 May 2019). (Photographed by Yong-Jun Lin).

### 3.2. District-Level Flood Wargame: Yonghe District

Table 2 shows the time slot for the wargame conducted in Yonghe District, which is the most populated district in New Taipei City. Some of the participants received training on how to use the FWGAP in advance. The results showed that combining the computer-aided and regular tabletop exercises could prove more effective [19]. For this wargame, the scenarios were printed out and distributed to the participants at the commencement of the game. Thus, they could use the FWGAP to evaluate the flooded area and look up relevant information.

The wargame process was initially orientated considering Yonghe's location in the southern part of New Taipei City. The primary setting for the scenario's construction entailed frontal contact with the northern part of New Taipei City at 6:00. By 10:00, the accumulated rainfall in Yonghe District had reached 180 mm. According to the CWB forecast, the frontal contact brought 350 mm of rainfall within six hours. The potential flooding map, based on 350 mm rainfall over six hours, was provided to the response team for reference (Figure 18).

**Table 2.** Time slot of flooding wargame conducted for Yonghe District.

| No. | Activities | Time (mins.) |
|---|---|---|
| 1 | The orientation process for the wargame and basic settings | 10 |
| 2 | Debriefing on the preparedness of the district for possible flooding due to torrential rain | 10 |
| 3 | Scenario 1/Debriefing 1 | 40 |
| 4 | Break | 15 |
| 5 | Scenario 2/Debriefing 2 | 40 |
| 6 | Special scenarios | 35 |
| 7 | Judge's observations and comments | 20 |

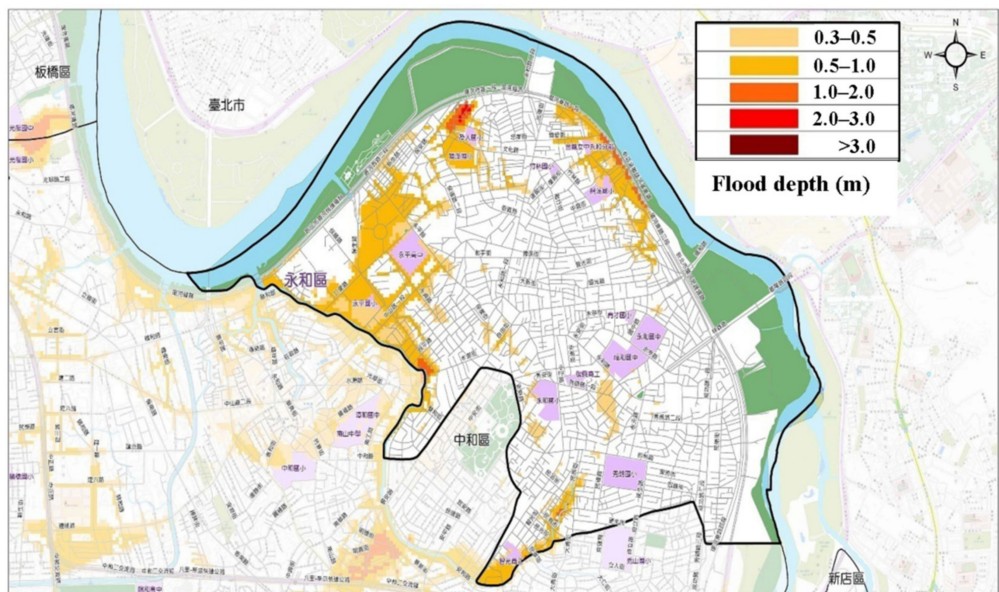

**Figure 18.** Flooding map based on 350 mm of rainfall for a duration of 6 h in Yonghe District.

The response team for Yonghe District conducted a debriefing of the flood preparedness plan. Subsequently, Scenario 1 was launched. The team discussed how to deal with the situation and the details of its response plan over a 40-min period. After a short break, Scenario 2 was initiated. Table 3 lists the main events and test functions relating to Scenarios 1 and 2. The events shown in Table 3 include the flooding event, evacuation, establishment of an incident command post, launch of the EOC, and power outages.

The judges of the wargame assigned two special scenarios, described below, to the response team, which used the FWGAP to assess the impacts and implement the necessary countermeasures.

**Special Scenario 1:**

At 13:00 on 30 July, according to images posted by the public on Facebook, drifting debris and broken bridge lights were observed stuck in the pipe supplying tap water upstream of the Yongfu Bridge. The water level of Xindian River was rising rapidly, leading to signs of rupture of the pipe, which also affected the safety of the bridge.

**Special Scenario 2:**

At 13:30 on 30 July, according to images posted by the public on Facebook, the flooding level at No. 24, Lane 39, Zhulin Road, Yonghe District had reached 1 m, and the flooding had expanded by nearly 400 m to the adjacent area. Taipower (the company supplying electricity) issued a statement projecting a period of at least two days to restore the electricity supply because of the equipment damage caused by the flooding. The news agencies reported that a care center for the elderly was located in the flooded area, and that many elderly residents with limited mobility required medical assistance because of the power outages.

The response team for Yonghe District used the FWGAP to conduct a thorough evaluation. Using this platform, the team was able to promptly locate and view the relevant positions of the debris and broken bridge lights mentioned in Scenario 1 (see the image to the right in Figure 19). They were also able to conduct a rapid assessment of the extent of the flooding and the affected population (see the image to the left in Figure 19). The pipe supplying tap water is visible in the image from Google Street View.

Table 3. Main events and test functions of the flooding wargame implemented for Yonghe District.

| Scheme. | Event | Time | Description | Test Function |
|---|---|---|---|---|
| 1 | A | 08:00 | The hourly rainfall is 30 mm. The EOC for New Taipei City requests the district office to launch its EOC at Level 2. | 1. CCTV monitoring of flooding situation<br>2. Field investigation<br>3. Launch of the district EOC |
| | B | 12:00 | The rainfall in nearby districts has crossed 50 (mm/h) over a 3-h period. The EOC for New Taipei City requests the district office to launch its EOC at Level 1. The director of the district office has to determine whether to announce a day off work in the afternoon. | 1. Reporting of the flooding events using the EMIS system<br>2. Field investigation<br>3. Day off announcement |
| | C | 12:35 | The accumulated rainfall in Yonghe District has reached 350 mm. There are large areas of flooding in the villages of Shanlin and Guilin. | 1. Traffic control<br>2. Voluntary evacuation |
| | D | 13:00 | The torrential rain has resulted in a rise in the water level of Xindian River, which is beginning to overflow into the area behind the dike around Yongfu Bridge. Five fishermen have been flushed into the river. | 1. Research and rescue<br>2. ICP |
| | E | 13:15 | As the water level in the Xindian River continues to rise, some disabled residents on the lower floors of buildings in Shanlin need to be evacuated. | 1. Evacuation of people on lower floors<br>2. Operation of shelters<br>3. Locating suitable shelters using the FWGAP |
| | F | 13:55 | The flooding depth of the Hyanho East Rd. reaches 1 m. | 1. Mandatory evacuation<br>2. Pumping of flood water<br>3. Organization of vehicles<br>4. Delivery of commodities<br>5. Estimate of the flood-affected people using the FWGAP |
| 2 | G | 15:00 | There are about 300 residents affected by the power outage along Showan and Sholain Streets. A tree has fallen down and hit the telecommunication box in No. 62, Mitzan Rd., thereby cutting off the fixed lines for 100 households.<br>The tap water pipe has burst in the location of a lamp post (no. 103342). | 1. Removal of fallen trees<br>2. Repair of power equipment<br>3. Repair of pipe supplying tap water<br>4. Repair of telecommunication equipment<br>5. Traffic Control |
| | H | 15:35 | The flooding depth at the intersection of Linshan Rd and Yonghen Rd is 0.5 m. The water level under Yongfu Bridge is less than 1 m. | 1. Moveable pumping station<br>2. Request for resources from the New Taipei City government<br>3. Traffic control at the bridge<br>4. Use of the FWGAP to estimate the flooded area |
| | I | 17:55 | The rainfall is stopping, and the EOC for New Taipei City is withdrawing the launch of its EOC. The recovery phase needs to be launched now. | 1. Pumping of water from the basements<br>2. Cleaning and disinfection<br>3. Subsidies provided to residents where there is significant flooding of their houses (a depth exceeding 0.5 m) |

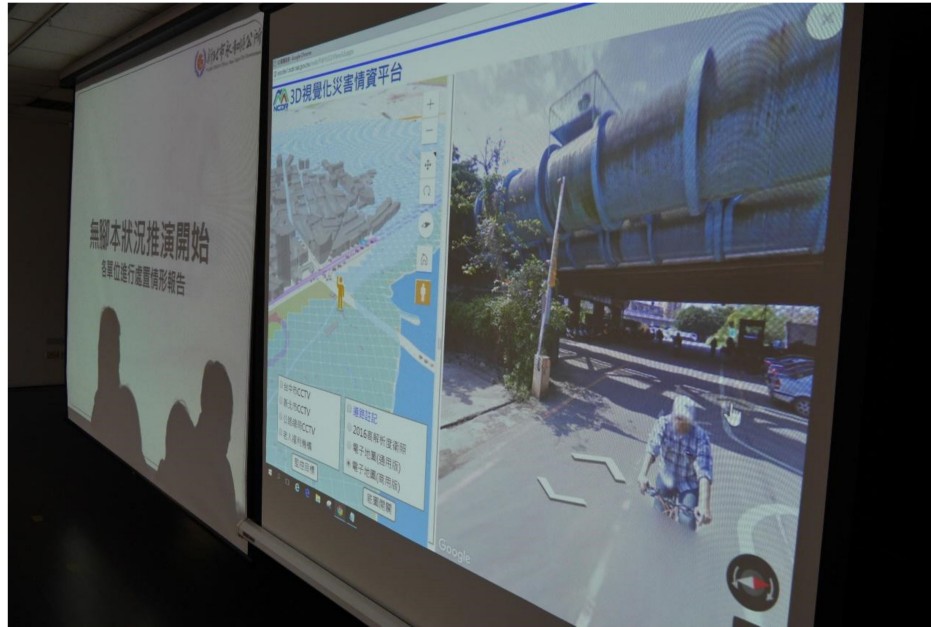

**Figure 19.** The tap water pipe in Yonghe District. (Photographed by Yong-Jun Lin).

In the case of Special Scenario 2, the FWGAP was used to evaluate the number of affected people (see the image to the left in Figure 19). The location and street view of a care center for the elderly was examined accordingly (central image in Figure 20 marked in green with the associated symbol). Using the street view function, the response team determined that this center was located on the second floor (in the image to the right in Figure 20). Elderly residents who required electricity for life support systems could be transported in high-chassis vehicles or boats across the floodwater and then transported by ambulance. If a CCTV was located nearby, the team could monitor its footage in real time to ascertain the flooding status and implement appropriate measures.

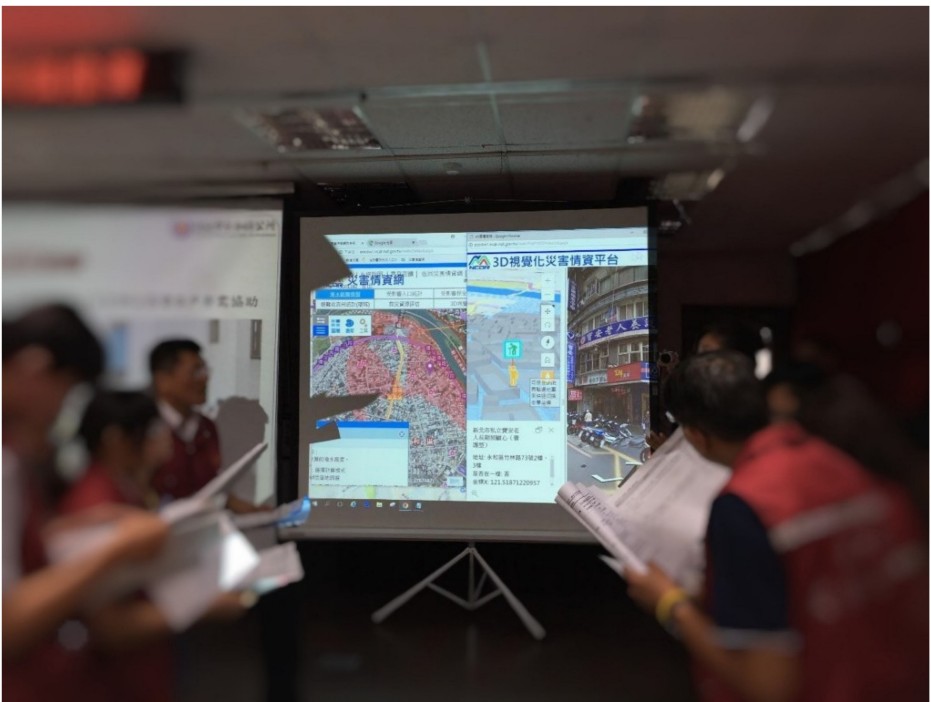

**Figure 20.** The street view of a care center for the elderly. (Photographed by Yong-Jun Lin).

After the wargame was concluded, the evaluators provided the response team with comments that are summarized below.

1.  The participants used the FWGAP to estimate the flooded area during events F and H and Special Scenarios 1 and 2. The affected population was estimated to be very large, but the entire population did not need to be evacuated. Vertical evacuation of residents on the lower floors was the first priority. The response team needed to think about how food could be delivered to those stranded on the upper floors.

2.  The response team used the FWGAP to obtain information about the care center for the elderly and nearby evacuation shelters. The capacity of the shelters should normally be evaluated annually, and the district office must issue orders for the shelters operations. In practice, the district office may open a high-priority shelter first, but it may not be the one nearest to the disaster spots. The FWGAP can provide information or inputs in this regard.

3.  Considering the extent of the flooding, the district does not have sufficient capacity to handle all of the events. Therefore, it could outsource some of this work to the New Taipei City government. The total requirement of human resources for handling the scenarios should be calculated during the process. For example, there were many flooded areas in this wargame and if each area required two police officers, the demand for police officers would exceed the supply capacity of the district office. In the observers opinions, this scenario was well beyond the capacities of the district.

4.  The pipe supplying tap water burst in Special Scenario 2. The response of the water company was that the pipe could be temporarily blocked for repair and the tap water diverted using alternative pipes. The participants tended to solve the problem immediately, but in reality, it could take days to repair such a large pipe that supplied water to thousands of residents not only in Yonghe District. In a worst-case scenario, the district office should establish some tap water supply stations to which water transported in water tanks could be delivered when needed.

*3.3. The Survey of the FWGAP*

An online survey was conducted to evaluate the FWGAP's fidelity and functionality. The survey also covered the differences between no-script and previously scripted wargames. Of the 16 participants, four (25%) were women and 12 (75%) were men. The majority of the participants (87.4%) belonged to the following age groups: 26–30 years, 31–35 years, and 36–40 years (Table 4), indicating that relatively young individuals with computer skills would use the FWGAP. Table 4 shows the participants years of work experience in the field of disaster prevention. Evidently, 75% of participants had 1–5 years of experience, indicating that they were young and relatively new to this field.

**Table 4.** The age and years of disaster-prevention work experience distribution of respondents who completed questionnaires.

| Age (Years) | 26–30 | 31–35 | 36–40 | 41–45 | 46–50 |
|---|---|---|---|---|---|
| Number of Participants | 5 | 5 | 4 | 1 | 1 |
| Percentage (%) | 31.2 | 31.2 | 25 | 6.2 | 6.2 |
| **Years of Disaster-Prevention Work Experience (Years)** | 1–5 | 6–10 | 11–15 | 15–20 | - |
| Number of Participants | 12 | 3 | 0 | 1 | - |
| Percentage (%) | 75 | 18.7 | 0 | 6.2 | - |

### 3.3.1. Fidelity

Figure 21 shows the results of the fidelity analysis. A five-point Likert scale was used for scoring responses: (1: "strongly disagree," 2: "disagree," 3: "undecided," 4: "agree," and 5: "strongly agree"). Approximately 38% of the participants thought that the FWGAP could simulate real flooding scenarios, whereas around 31% of the participants disagreed (Q1, Figure 21a). The respondents were asked whether three of the FWGAP's functions, namely, the three-dimensional buildings, Google Street View map, and CCTVs could increase the fidelity of the simulation. Approximately 44% of the participants agreed that three-dimensional buildings could increase fidelity (Q2, Figure 21b). About 56% of participants agreed that three-dimensional buildings could increase fidelity, and about 19% of participants strongly agreed that three-dimensional buildings could increase fidelity (Q3, Figure 21c). In the case of CCTV, 38% of the participants agreed it could increase fidelity, and 19% strongly agreed with this statement (Q4, Figure 21d). Figure 21e depicts a radar chart of the three functions. The weighted points indicate that Google Street View map was ranked highest (3.81), followed by CCTV (3.44), with three-dimensional buildings being ranked lowest (3.31). The weighted points for all three functions exceeded the average point value (3.00), but the difference was not significant. This result indicates that there is considerable scope for improving these functions.

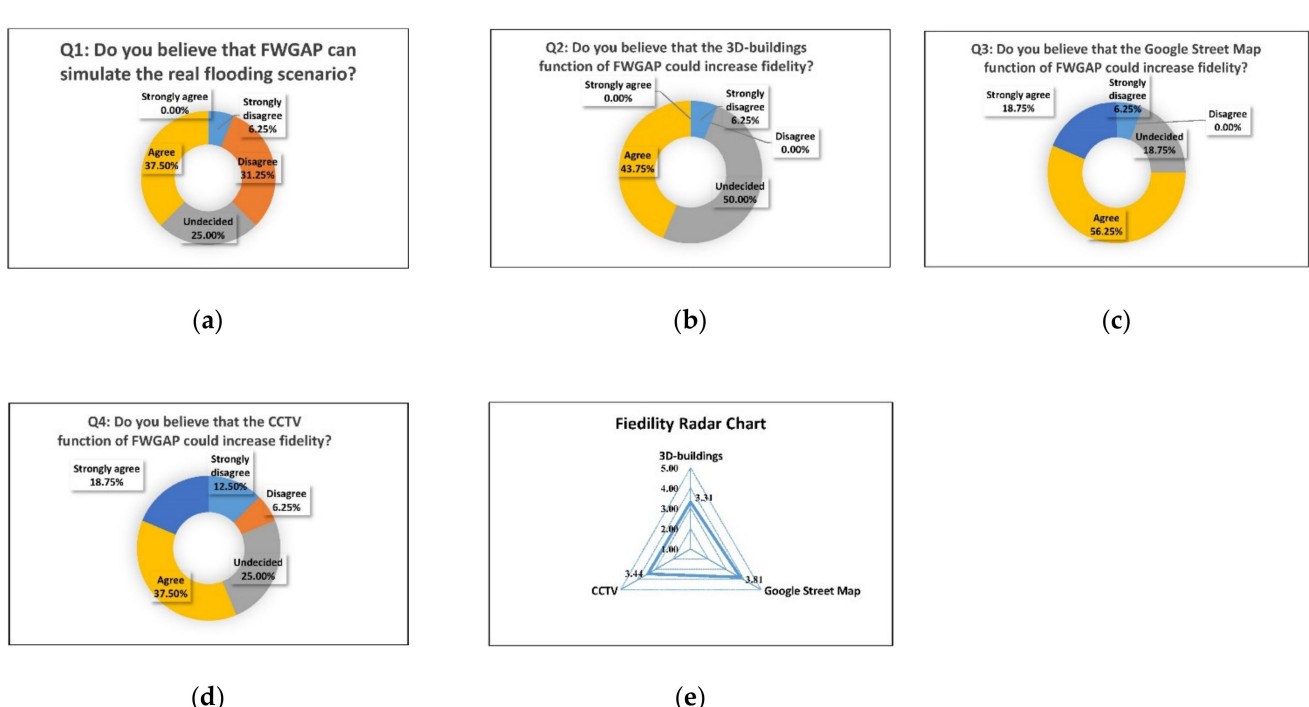

**Figure 21.** Assessment of the simulation's fidelity: (**a**) real flood scenario simulation, (**b**) 3D-buildings, (**c**) Google Map, (**d**) CCTV, and (**e**) radar chart.

### 3.3.2. Functionality

The FWGAP has functions for estimating the flooded area and the affected and vulnerable populations, and for searching for nearby facilities such as hospitals, fire stations, and evacuation shelters. Besides, the FWGAP's "little painter" function could identify resources that could facilitate the debriefing and aid the wargame participant's decision making through access to images of spatialized scenarios and resources displayed on the map. The results show that 56.25% of the participants agreed that all of the functions were effective (Q5, Figure 22a, Q7, Figure 22c, Q10, Figure 22f, Q11, Figure 22g, Q12, Figure 22h). About 44% of the participants believed that the flooding area estimated with the FWGAP was larger than anticipated (Q6, Figure 22b). Moreover, about 31% of the

participants believed that the affected population estimated with the FWGAP was also larger than expected (Q8, Figure 22d). When asked if the evacuees should be evacuated to higher levels in the event of flooding, 68.75% of the participants answered "yes," and 25% of participants answered "no" (Q9, Figure 22e). Surprisingly, about one-fourth of the participants did not think that evacuees should be evacuated vertically during a flooding event. Approximately 37.5% of the participants strongly agreed that the search function was effective (Q11, Figure 22g). About 31% of the participants strongly agreed that "the little painter" could help to mark the resources or facilitate the debriefing (Q12, Figure 22h). Figure 22i depicts a radar chart of the five functions. The weighted points for the "search function" were highest (3.81), followed by the little painter (3.69), with all other functions having 3.44 points. Thus, the weighted points for all of the functions were above 3.

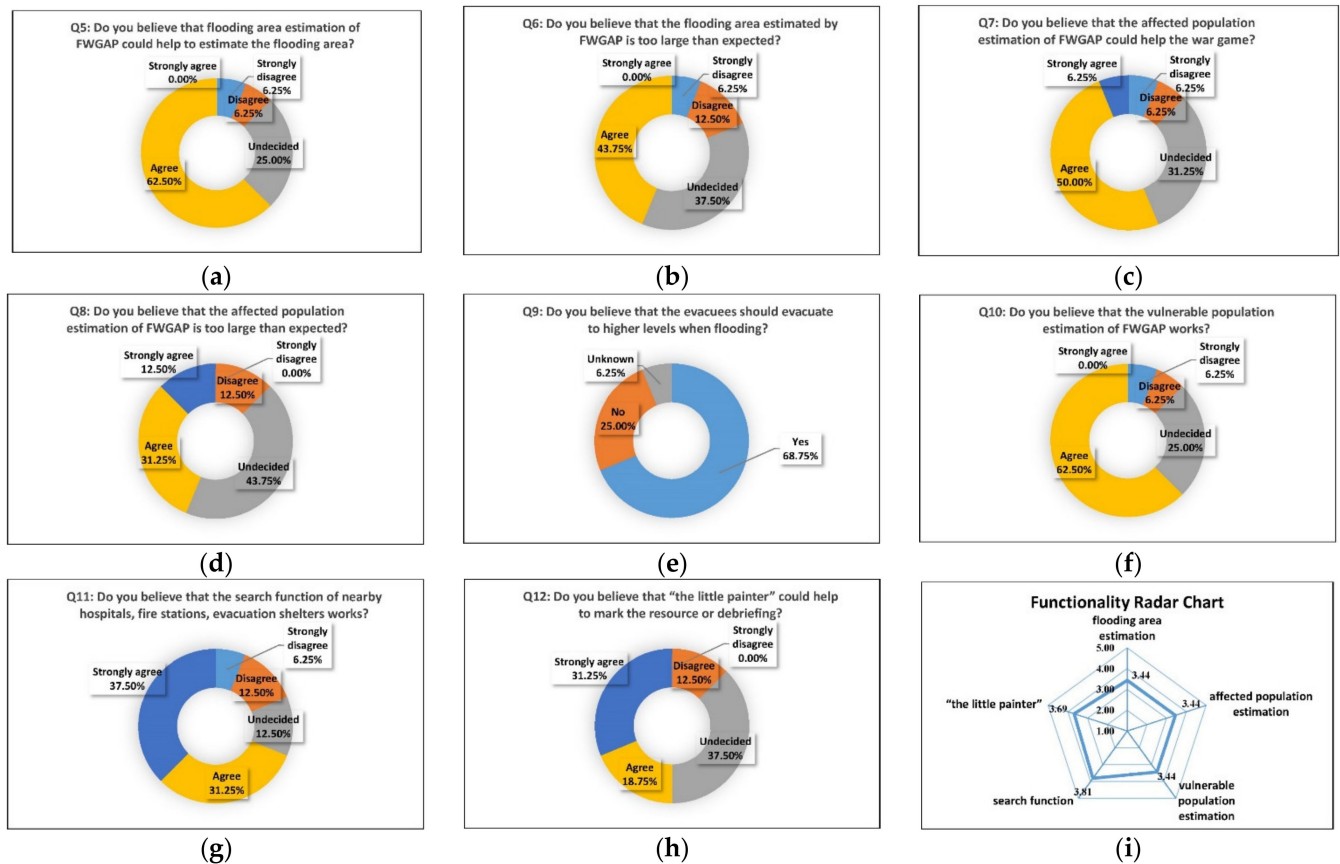

**Figure 22.** Assessment of the platform's functionality: (**a**) flooded area estimation, (**b**) if estimated area is larger than expected, (**c**) affected population estimation, (**d**) if estimated, affected population is larger than expected, (**e**) evacuees evacuate to higher ground, (**f**) vulnerable population estimation, (**g**) search function, (**h**) little painter, and (**i**) radar chart.

### 3.3.3. Ease of Use

If a function is not easy to use, the users may not use it. To ascertain the ease of use, eight functions were evaluated: the flooding area estimation; the affected population estimation; the vulnerable population estimation; the search function for locating nearby hospitals, fire stations, and evacuation shelters; three-dimensional buildings; Google Street View maps; CCTV; and "the little painter." Figure 23 depicts a radar chart. Among these functions, the Google Street View function was ranked the highest with 3.63 points, followed by the search function with 3.5 points.

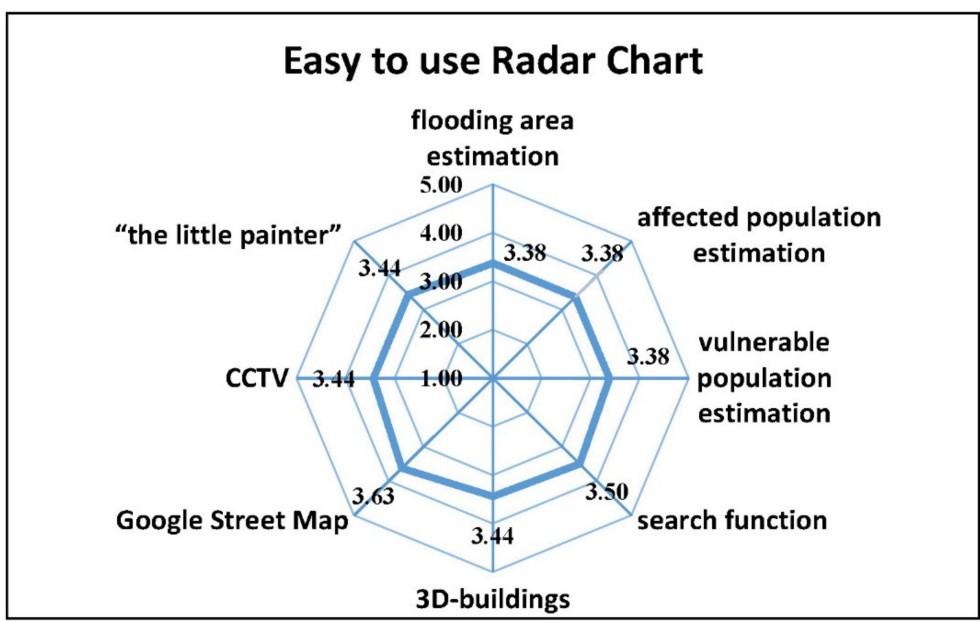

**Figure 23.** Assessment of the ease of use of the platform's functions, based on the results of the questionnaire survey.

### 3.3.4. The Difference between No-Script and Previously Scripted Wargames

About 56% of the participants agreed that the FWGAP provides background information that helps in decision making during wargames (Q14, Figure 24a). Over 62% of participants agreed that no-script wargames helped them to identify problems in the SOPs (Q15, Figure 24b). Sixty-two percent of the participants believed that the scenarios in the no-script game exceeded the capacities of the district offices (Q16, Figure 24c). From these results, it can be concluded that the participants considered tackling the scenario to be well beyond their capacities. Half of them agreed that the no-script wargame could strengthen horizontal coordination among departments (Q17, Figure 24d). The results suggest that there are few situations within these scenarios for them to collaborate horizontally or that lack of collaboration may be due to the prevailing administrative systems in Taiwan, where each department is assigned a distinct area of responsibility and is accordingly held responsible for any failure in measures it adopts in response to a particular scenario. About 56% of the participants shared the view that the no-script wargame differs from the previous scripted wargame (Q18, Figure 24e). Figure 24f shows the weighted points for those questions, revealing that the scenarios of the no-script wargame exceeded the capacity of the district offices (weighted points: 3.88).

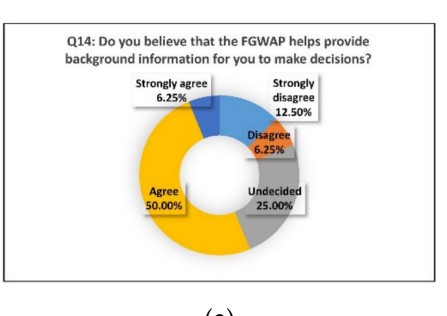

(a)

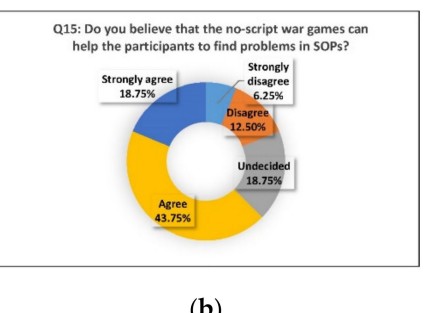

(b)

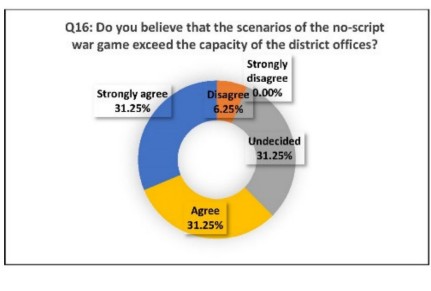

(c)

**Figure 24.** *Cont.*

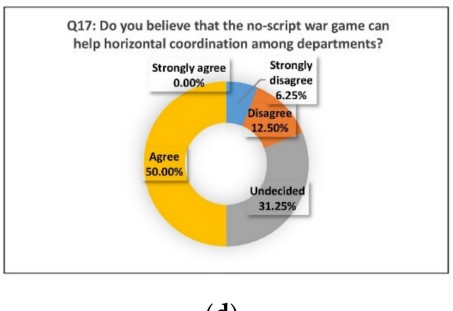
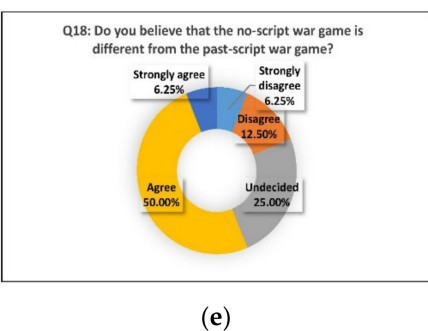
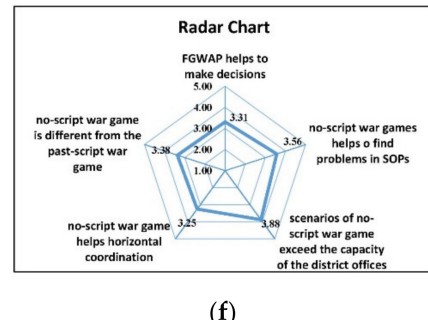

(**d**)           (**e**)           (**f**)

**Figure 24.** Assessment of the differences between no-script and previously scripted wargames: (**a**) background information, (**b**) problems in SOP, (**c**) capacity exceedance, (**d**) horizontal coordination, (**e**) difference between no-script wargame and script wargame, and (**f**) radar chart.

### 3.3.5. Discussion

1.  A few individuals stated that the estimate for the affected population was larger than expected. The survey results show that about 56% of the participants considered the estimation of the affected population using the FWGAP to be much larger than expected (Q7, Figure 21c). The estimate was based on the minimum statistical unit, which sometimes resulted in the platform-estimated population being more significant than that of the estimated flood-affected area. The minimum statistical unit is used in Taiwan to ensure accurate population estimates. The rationale is that if one minimum statistical unit intersects with a flooded grid unit, the total population of that minimum statistical unit would be added to the total population of the affected area. A possible solution is to calculate the proportion of flooded grids to that of the area of the minimum statistical unit. Another solution is to apply big data analysis to cellular users signals to estimate the population at different times, such as during the day and at night. However, obtaining operational data for all cellular phones is challenging. Moreover, some people do not use cellular phones at all. During the COVID-19 pandemic in 2020, the NCDR used data obtained from the largest cellular phone operator to estimate the population at the sighting spots for man flow control. Our findings indicate that some of the wargame participants assumed that all of the affected people needed to be evacuated. The number of affected people usually exceeds the number requiring evacuation. In flooded areas, building residents should be evacuated vertically to higher levels. Only disabled or elderly residents on the first floors of buildings in the flooded area should be evacuated first with the assistance of the rescue teams. For others, vertical evacuation is a better option, and it is hazardous to perform horizontal evacuations when floodwaters are still rising.

2.  The survey results revealed that 44% of participants believed that the flooded area estimated with the FWGAP was larger than expected. Such an estimation could be made after determining the center, flooding depth, and distance of impact using the platform. Because the water level at the center usually spreads horizontally to nearby elevation grids, the flooding area can be determined. This method is effective for simulating large-scale flooding events with water depths greater than 1 m. In this situation, the drainage system does not drain the floodwater efficiently, leaving the conduits full of water. Gravity then causes an overland flow of the water to low-lying areas. The platform enables a rapid estimation of the population affected by such massive flooding. Chen et al. [39], argued that most flood simulations focus on regular events. Therefore, simulations should attend more to crisis flooding events [40–42]. In addition to its use in estimations, the platform also provides information with a high degree of accuracy about the flooded areas during historical flooding events and the flooding potential within different scenarios, using a hydraulic model to simulate the drainage system. Observations of the wargame revealed that given the ease of use of the depth-setting method, the participants seldom used the two types of information

mentioned above. Moreover, hydraulic modeling of overland flow using fine grids takes hours to complete. In the future, massive volumes of pre-run overland flow simulations with multiple rainfall scenarios and artificial intelligence technology may need to be used to develop rapid and accurate estimates of flooded areas. Data on flooded areas during historical events are also essential for the calibration of overland flow simulations. In recent years, some historical flooding spots have been compiled by the NTC. However, only information on flood locations and depth is available, with the areas flooded only available for a few spots. Different flooding areas during various events may have the same flooding depth. Consequently, following the adoption of this platform for conducting flooding wargames, the authorities should request more information about the flooded area from citizens calling to report floods.

3. To facilitate the dispatch of available resources, the participants hope to be able to add frequently used icons, such as the incident command post (ICP), the traffic control point (TCP), the evacuation area and the deployed personnel. Accordingly, a new function called the "little painter" has been incorporated into the platform, allowing users to draw and label objects on the platform. Figure 25 illustrates the labeling of an ICP, a temporary gathering point (TGP) and a TCP for facilitating reporting in a wargame.

   These functions enable the participants to visualize the distribution of resources. In the past, participants appeared to have unlimited resources at hand that they could dispatch as they pleased within a short time. In reality however, resources are limited and flooded roads may impede their dispatch. Following the recommendation of one of the surveyed respondents to "add the frequently used icons, such as the field command post, to the little painter function", this function will be added to an updated version of the FWGAP.

4. Another recommendation from the users was to categorize the flooding depth into different levels, using different colors to facilitate reading. This function has been added in the latest version of the platform. Moreover, the flooding depth meets the regulations of the Water Resources Agency, Taiwan.

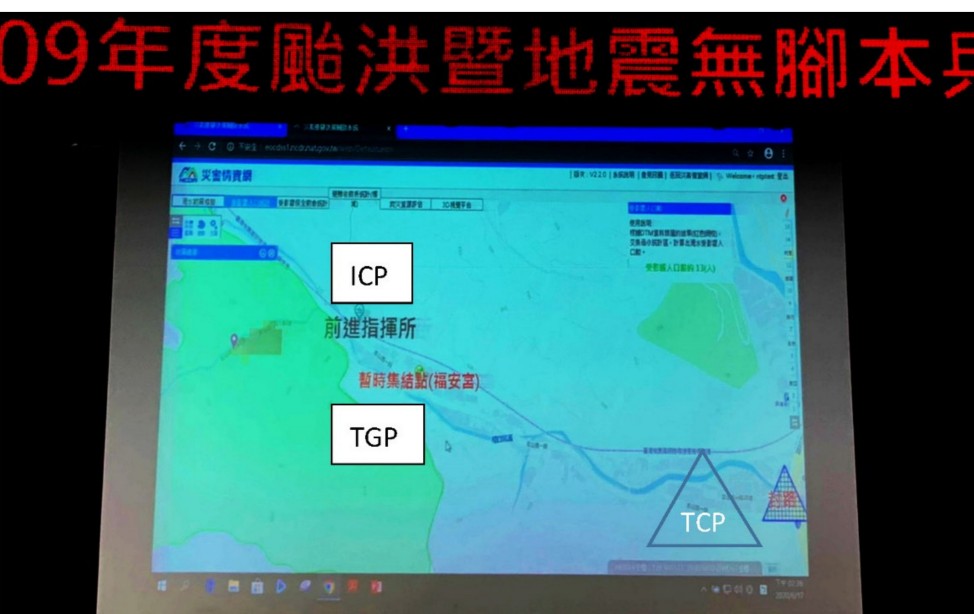

**Figure 25.** Easy-to-read labeling of the FWGAP. (Photographed by Yong-Jun Lin).

### 4. Conclusions and Future Work

1.  GIS, CCTV images, 3D buildings, and Google Street View functions have been incorporated within the FWGAP. Three methods can be used to estimate the flooded area: (1) the flooding center and depth with a DTM, (2) historical flooding spots and (3) potential flooding maps.

2.  The estimated affected areas can provide the basis for a spatial analysis to locate nearby resources such as shelters, disaster relief resources, hospitals, and care centers for the elderly, as well as the affected population. The three-dimensional building and street view functions provide the response team with an immersive experience: for example, they can view the floors of a care center for the elderly. The CCTV's real-time images enable instant surveillance of the flooding situation. The FWGAP can be used not only in wargames, but also during disasters.

3.  New layers and modules can easily be added to the platform given its flexibility. The participants can use the standard platform when discussing the response plan. The survey results revealed that the primary users of the platform are relatively young. Over half of the participants thought that three-dimensional buildings, Google Street View maps, and CCTV images could increase the fidelity of the simulation. However, 44% and 56% of the participants, respectively, felt that the estimates of flooded areas and affected populations were too large. The search function for locating resources near the flooded area was considered the most useful function. There is considerable scope for improving these two functions. The "little painter" function, which enabled participants to label the ICP, control area, and traffic control spots, was ranked second for functionality.

4.  About 56% of the participants felt that the FGWAP would provide useful background information during wargames. About half of the participants believed that a no-script flooding wargame using the FWGAP could facilitate the identification of issues relating to the SOPs and promote horizontal coordination among departments. In conclusion, the FWGAP was found to be a useful platform for a no-script flooding wargame.

5.  Currently, the system can only be applied with pre-set scenarios that have been constructed based on 24-h quantitative rainfall flooding, potential maps, or historical flooding spots. In the future, it will be possible to use multiple sets of pre-calculated flooding potential maps, for example, a 24-h 350 mm flooding potential scenario, which can be rapidly generated through the interpolation of a 24-h 300 mm flooding potential map and a 24-h 400 mm flooding potential map. Thus, the scenario's physical settings would be generated automatically. In addition, new high-spatial resolution flooding potential maps, which takes into account the building block effect will be adopted in the platform for higher fidelity.

6.  The FWGAP can query the locations and capacities of nearby evacuation shelters. However, it cannot provide information on the actual deployment of resources. For example, Shelter A, which has the capacity to accommodate 300 people, now has 200 evacuees. More metadata on three-dimensional buildings could be added in the future. For example, the FWGAP metadata on care centers for the elderly currently only includes their names and addresses. Therefore, metadata on, for example, their capacities and numbers of floors could be added to the database.

**Author Contributions:** Conceptualization, W.-R.S. and Y.-J.L.; methodology, W.-R.S. and Y.-J.L.; software, C.-H.H. and C.-H.Y.; formal analysis, Y.-J.L. and C.-H.H.; investigation, C.-H.H.; resources, Y.-F.T.; data curation, C.-H.H., C.-H.Y. and Y.-F.T.; writing—original draft preparation, W.-R.S. and Y.-J.L.; writing—review and editing, W.-R.S., Y.-J.L. and Y.-F.T.; visualization, C.-H.H. and C.-H.Y.; supervision, W.-R.S., Y.-J.L. and Y.-F.T.; project administration, W.-R.S., Y.-J.L. and Y.-F.T.; funding acquisition, Y.-J.L. All authors have read and agreed to the published version of the manuscript.

**Funding:** This research was funded by the Ministry of Science and Technology in Taiwan, grant number 109-2119-M-002-018.

**Institutional Review Board Statement:** The study was conducted according to the guidelines of the Declaration of Helsinki, and approved by the National Science and Technology Center Research Ethics Committee (3-2NCDR-PR-002; date of approval: 15 May 2019).

**Informed Consent Statement:** Informed consent was obtained from all subjects involved in the study.

**Data Availability Statement:** Not applicable.

**Acknowledgments:** The authors acknowledge the data provided by the New Taipei City government, Taiwan.

**Conflicts of Interest:** The authors declare that they have no competing interest.

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
