# Peer review of "3D GIS Platform for Flood Wargame: A Case Study of New Taipei City, Taiwan"

_water, doi:10.3390/w13162211_

Round 1
Reviewer 1 Report
Some minor issues:
- I tend to prefer "wargame" to "war game" (either is OK)
- Line 97: "printed" (not "print-out")
- Line 112: give link for Stop Disasters game? https://www.stopdisastersgame.org/
- outline starting at line 270 needs some formatting (indent each description entirely) (align the indents, too)
- 331: maybe it's the manuscript system for reviews, but I see a whole bunch of Enter characters (↵) in Figure 3
- Figure 19: do we need permissions for those whose faces are visible? You might need to blur them out
- Figures 20, 21, 23: please spread out the figures
- Line 658: lots of debate if we should just say "center" and not "epicenter" https://www.politico.com/news/magazine/2020/03/28/coronavirus-united-states-epicenter-new-york-152716
- rename Line 705 to Conclusions and Future Work
Reviewer 2 Report
The authors implement/create a GIS tool to evaluate the possible flooded areas starting from the different initial conditions and by using different qualitative and quantitative computational approaches. This processing tool is part of an interessant Flood War Game Platform.
The paper is well written, and it is appreciable as the authors focused to make reader understand the importance of this type of approach to disaster management/reduction.
In principle, I might as well agree with them, but I believe that the target reader of this journal expects a more scientific approach to the problem.
In primis, the introduction is really heavy. This is unbalanced concerning the rest of the work. Line 44 to line 82 is a disquisition on the Governative strategy on the disaster that can be compressed into 5 -10 rows.
please, is opportune to split the introduction into two-part: 1) on the game simulating strategy to natural risk mitigation in urban areas; 2) stat of the art respect other flood game tools.
In addition in the introduction, it needs that the authors explain as the simplified GIS-game-tool gives a good representation of the real scenarios than the physical simulation models applied that consider the regional flood hazard and the local susceptibility. In plaining practice, this latter comes from a study based on assessing the fragilities of natural/urban systems.
page. 9: What are the theoretical bases that induce the hypothesis that flood is produced by increasing the water level around a designated center point?
pag. 11 Map of the potential flooded area does take into account the building elements in the propagation?
page.13: Who are the decision-makers?
table 4: I don't understand the last two rows, seem the same as the previous ones but with different values.
Page 23 and 24. In scientific works, the performance of a model, frequently named reliability, is estimated by comparison on experimental cases or in back-analysis on real events. In this case, the performance of the proposed tool is demanded by the "fidelity in the representation of the scenery" and by "functionality". These qualitative subjects are, here, provided to the opinions of the qualified users. But an opinion remains an opinion, however; it is very difficult to admit this kind of meter to evaluate the efficacy of the introduced tool.
Author Response
Please see the attachment.

This manuscript is a resubmission of an earlier submission. The following is a list of the peer review reports and author responses from that submission.
Round 1
Reviewer 1 Report
The novelty/originality is not justified to highlight that the manuscript contains sufficient contributions to the new body of knowledge. The knowledge gap needs to be further addressed. Also, structure of this manuscript is out of an academic writing.
Reviewer 2 Report
A BRIEF SUMMARY
The paper titled “An Assistance Platform for a GIS-Based Flood War Game: A Case Study of New Taipei City, Taiwan” presents a good topic for readers of this Journal. Hovewer, several lacks emerge after reading the paper. Below is the list of serious lacks to justify my decision to reject the manuscript.
- Lack of novelty. This manuscript represents a technical report of a case study.
- Lack of an adequate method description.
- Results are presented in a confusing way.
- Too little bibliography for this type of work, on a so broad topic. I strongly suggest that the authors try to add some more references especially in the "part 1 (introduction)" of the paper to make the foundation for the arguments stronger.
- Lack of conclusions. The conclusions section reports a summary of the results. Where are conclusive considerations on the presented work? You have to highlight novelty of this paper and potential future development.
- There are too much figures for a scientific paper. For example, you have to delete Figure 16-18-19-24.